# ATOMMOF: All-Atom Flow Matching for MOF-Adsorbate Structure Prediction

## Abstract

Deep generative models have shown promise for modeling metal-organic frameworks (MOFs), but existing approaches (1) rely on coarse-grained representations that assume fixed bond lengths and angles, and (2) neglect the MOF-adsorbate interactions, which are critical for downstream applications. We introduce ATOMMOF, a scalable flow-based model built on an all-atom Diffusion Transformer that maps 2D molecular graphs of building blocks and adsorbates directly to equilibrium 3D structures without imposing structural constraints. We further present scaling laws for porous crystal generation, indicating predictable performance gains with increased model capacity, and introduce Feynman-Kac steering guided by machine-learned interatomic potentials to improve geometric validity and sampling stability. On the (MOF-only) BW dataset, ATOMMOF increases the match rate by 35.0% and reduces RMSE by 32.64%. On the ODAC25 dataset (MOF-adsorbate), ATOMMOF is substantially more sample-efficient than grand canonical Monte Carlo in recovering adsorption configurations and can identify candidates with lower adsorption energies than the reference dataset.

## 1. Introduction

Metal-organic frameworks (MOFs) are porous, crystalline materials whose highly tunable chemistries and large surface areas make them central to gas separations (Qian et al., 2020), catalysis (Lee et al., 2009), and sensing (Kreno et al., 2012). One particularly urgent application is direct air capture (DAC), in which $CO_2$ is selectively adsorbed from ambient air. This task is intrinsically difficult as $CO_2$ is extremely dilute and atmospheric species such as $H_2O$ compete for adsorption sites (Bose et al., 2024). Accurate prediction of

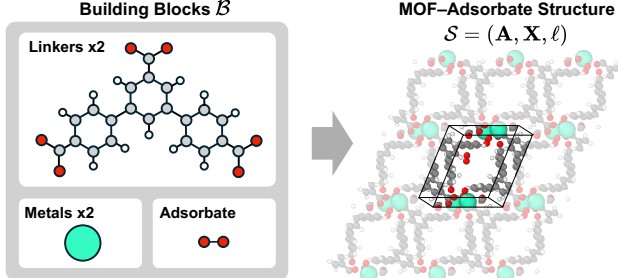

*Figure 1.* **MOF-adsorbate structure prediction.** Given a set of building blocks $\mathcal{B}$ (comprising linkers, metal nodes, and adsorbate identity), the goal is to predict the equilibrium structure $\mathcal{S} = (\mathbf{A}, \mathbf{X}, \ell)$, defined by atomic numbers $\mathbf{A}$, Cartesian coordinates $\mathbf{X}$, and lattice parameters $\ell$.

host–guest interactions in MOF structures is a critical step in discovering effective materials for carbon capture.

Traditional computational approaches rely on simulation-based workflows, such as grand canonical Monte Carlo (GCMC; Adams, 1975) and density functional theory (DFT; Kohn & Sham, 1965), to sample adsorbate positions and relax the system to a local energy minimum by force field-based iterative minimization. While accurate, these methods are prohibitively expensive for MOFs, which often contain hundreds of atoms in a unit cell. Furthermore, they face a fundamental challenge: accurate screening requires a known 3D structure of the host MOF. This restricts screening to existing databases, while the space of hypothetical MOFs is vast, given the diverse nature of building blocks.

To overcome these challenges, deep generative models have emerged as a promising alternative. However, existing models suffer from two limitations. First, they often operate on a coarse-grained resolution with rigid-body assumptions (Kim et al., 2025b; Jiao et al., 2025; Pan et al., 2026) or fixed bond lengths and angles (Kim et al., 2025a). While this simplifies the modeling task, it fails to capture continuous variations in bond lengths, bond angles, and coordination geometries that arise from local chemical environments and host–guest interactions. Furthermore, all the works generate the host structure in isolation, neglecting the adsorbate interactions that ultimately dictate performance in real-world applications like DAC.

**Contribution.** We introduce ATOMMOF, a scalable flow-matching model based on a unified Diffusion Trans-

[1]Anonymous Institution, Anonymous City, Anonymous Region, Anonymous Country. Correspondence to: Anonymous Author <anon.email@domain.com>.

Preliminary work. Under review by the International Conference on Machine Learning (ICML). Do not distribute.

former (DiT; Peebles & Xie 2023) architecture that solves the all-atom MOF-adsorbate structure prediction problem. Unlike prior work, ATOMMOF operates in a fully unconstrained space, mapping discrete building block identities and adsorbate types, e.g., $CO_2$, $H_2O$, directly to equilibrium three-dimensional atomic coordinates and lattice parameters. By relaxing rigid constraints, we enable the modeling of flexible structures with high accuracy.

In addition, we introduce two key technical innovations. First, we demonstrate that scaling laws hold for MOF generation, showing that performance improves predictably with model size (Figure 4). Next, we propose Feynman-Kac steering with machine learning interatomic potentials (MLIP), which significantly improves the geometric validity and stability of the generated structures.

Our contributions are summarized as follows:

- **All-atom generative framework:** We propose the first all-atom model for MOF structure prediction. We demonstrate the necessity of relaxing rigid-body assumptions to capture realistic structural variations and validate our approach on the BW dataset (Boyd et al., 2019), improving match rate by over $35.0\%$. We also establish the first scaling laws for porous crystal generation, suggesting that further performance gains are attainable through continued scaling. Finally, to ensure geometric validity in this unconstrained space, we introduce Feynman-Kac steering with MLIP, which improves validity by $10.86\%$ and stability by $84.19\%$.

- **MOF-adsorbate structure prediction:** We formulate the task of MOF-adsorbate prediction. Given only 2D molecular graphs of building blocks and an adsorbate identity, ATOMMOF predicts the structure of the MOF-adsorbate complex. On the ODAC25 dataset (Sriram et al., 2025), we demonstrate that our model generates diverse adsorption configurations with substantially less wall-clock time than GCMC. We also show that initializing MLIP relaxation with ATOMMOF samples discover adsorption configurations with adsorption energies below the lowest energy configurations in the reference dataset.

## 2. Related Work

### 2.1. Machine Learning for Crystal Structure Prediction

Most existing work on crystal structure prediction has focused on inorganic crystals. DiffCSP (Jiao et al., 2024a) pioneered this direction by jointly denoising lattice and coordinate information using a periodic E(3)-equivariant architecture. Subsequent works have refined this approach by better handling symmetries: EquiCSP (Lin et al., 2024) accounts for lattice permutation symmetry, while DiffCSP++ (Jiao et al., 2024b) and SymmCD incorporate space group constraints. Beyond diffusion, other generative frameworks have also been explored; FlowMM (Miller et al., 2024) uses equivariant flow matching, CrysBFN (Wu et al., 2025) applies a periodic E(3) Bayesian flow network, and OMatG (Höllmer et al., 2025) adapts the stochastic interpolant framework to further enhance performance.

Unlike inorganic crystals, structure prediction for MOFs presents unique challenges due to their modular nature and larger system size. Existing methods manage this complexity with the rigid body assumption: MOFFlow (Kim et al., 2025b) predicts building block rotations and translations via Riemannian flow matching, MOF-BFN (Jiao et al., 2025) operates on fractional coordinates to enforce periodicity, and MOF-LLM (Pan et al., 2026) proposed progressive training with a large language model. While MOFFlow-2 (Kim et al., 2025a) mitigates this by incorporating torsion angles around rotatable bonds, it still relies on the assumption that local geometries (bond lengths and angles) generated by cheminformatics tools are fixed. In this work, we introduce an all-atom model that relaxes these constraints, enabling fully flexible structure prediction.

### 2.2. Generative Model for Molecular Complexes

A closely related problem to material structure prediction is protein folding. While early works like AlphaFold 1 and 2 (Senior et al., 2020; Jumper et al., 2021) focused on predicting isolated protein structures, AlphaFold 3 (Abramson et al., 2024) advanced the field by predicting the joint structures of complexes, including proteins, nucleic acids, and small molecules. Recent works such as Boltz (Wohlwend et al., 2025; Passaro et al., 2025), Chai-1 (team et al., 2024), Protenix (Team et al., 2025), and SeedFold (Yi et al., 2025) continue to model molecular complexes.

Similarly, in heterogeneous catalysis, generative models explicitly capture catalyst-adsorbate interactions (Kolluru & Kitchin, 2024; Anonymous, 2026). We extend MOF structure prediction along these lines by enabling the joint prediction of the full MOF-adsorbate system.

## 3. Method

### 3.1. Problem Formulation

In this work, we consider structure prediction of MOFs. To be specific, we predict atomic coordinates of an MOF-adsorbate system given the underlying molecular graph for the building blocks (organic linkers and metal ions) and optionally adsorbates (Figure 1).

**Crystal representation.** A MOF structure $\mathcal{S}$ is specified by a unit cell that repeats on a three-dimensional lattice. We represent a unit cell with $N$ atoms as $\mathcal{S} = (\mathbf{A}, \mathbf{X}, \boldsymbol{\ell})$, where $\mathbf{A} = \{a_i\}_{i=1}^{N} \in \mathcal{A}^N$ are the atom types (with $\mathcal{A}$ the

set of chemical elements) and $\mathbf{X} = \{x_i\}_{i=1}^N \in \mathbb{R}^{N \times 3}$ are the atomic coordinates. The lattice is given by a matrix $L = (l_1, l_2, l_3) \in \mathbb{R}^{3 \times 3}$ whose columns are lattice vectors. We define $\boldsymbol{\ell} = (\boldsymbol{d}, \boldsymbol{\phi})$ as a reparametrization of the lattice matrix, where $\boldsymbol{d} = (a, b, c) \in \mathbb{R}_+^3$ are the edge lengths and $\boldsymbol{\phi} = (\alpha, \beta, \gamma) \in [0°, 180°]^3$ are the inter-edge angles. The infinite periodic crystal associated with $\mathcal{S}$ is $\{(a_n, x_n + Lm) \mid n \in [1, N], m \in \mathbb{Z}^3\}$ where $M = (m_1, m_2, m_3)^\top$ is a 3D translation of $L$.

**Building block representation.** The MOF-adsorbate system can be represented as a union of molecular graphs, $\mathcal{B} = \mathcal{B}_{\text{MOF}} \cup \mathcal{B}_{\text{ads}}$. The framework consists of $M$ building blocks $\mathcal{B}_{\text{MOF}} = \{\mathcal{G}_i\}_{i=1}^M$, where $\mathcal{G}_i$ denotes the $i$-th molecular graph consisting of atoms as vertices and bonds as edges. Similarly, the adsorbates are denoted by $\mathcal{B}_{\text{ads}} = \{\mathcal{H}_j\}_{j=1}^P$, where $P = 0$ for the bare MOFs and $\mathcal{H}_j$ denotes the molecular graph of $j$-th adsorbate.

### 3.2. Variational Flow Matching for MOFs

In this work, we adopt variational flow matching (VFM; Eijkelboom et al., 2024), which allows us to train with an $L^1$ objective and empirically improves convergence. Concretely, VFM is a generative modeling framework that learns to transform samples from a prior distribution $p_0(\mathbf{z}_0)$ to a target data distribution $q$ by predicting clean data from noisy observations, given the building blocks $\mathcal{B}$.

**Conditional probability path.** We apply VFM to jointly model the atomic coordinates $\mathbf{X}$ and lattice parameters $\boldsymbol{\ell}$; for notational simplicity, we use a generic variable $\mathbf{z} = (\mathbf{X}, \boldsymbol{\ell}) \in \mathbb{R}^{N \times 3} \times \mathbb{R}^6$. To be specific, we define a conditional probability path $p_t(\mathbf{z} \mid \mathbf{z}_1)$ that interpolates between the prior and target.[1] At $t = 0$, we sample from the prior $\mathbf{z}_0 \sim p_0$, and at $t = 1$, we recover the data $\mathbf{z}_1 \sim q$. We use the linear interpolation path:

$$\mathbf{z}_t = (1 - t)\mathbf{z}_0 + t\mathbf{z}_1, \quad \mathbf{z}_0 \sim p_0, \ \mathbf{z}_1 \sim q. \quad (1)$$

**Training and inference.** Given a noisy sample $\mathbf{z}_t$, VFM learns a variational posterior $q_t^\theta(\mathbf{z}_1 \mid \mathbf{z}_t)$ that approximates the true posterior $p_{1|t}(\mathbf{z}_1 \mid \mathbf{z}_t)$. The VFM objective minimizes the expected negative log-likelihood:

$$\mathcal{L}_{\text{VFM}}(\theta) = -\mathbb{E}_{t, \mathbf{z}_t, \mathbf{z}_1}\big[\log q_t^\theta(\mathbf{z}_1 \mid \mathbf{z}_t)\big], \quad (2)$$

where $t \sim \mathcal{U}[0, 1]$, $\mathbf{z}_1 \sim q$, and $\mathbf{z}_t \sim p_t(\mathbf{z} \mid \mathbf{z}_1)$. At inference time, we generate samples by solving the ordinary differential equation (ODE) $d\mathbf{z}_t = \mathbf{v}_t(\mathbf{z}_t)dt$ defined by the velocity field $\mathbf{v}_t(\mathbf{z}_t) = (\boldsymbol{\mu}_t^\theta(\mathbf{z}_t) - \mathbf{z}_t)/(1 - t)$.

**Parameterization.** We model the variational posterior $q_t^\theta(\mathbf{X}_1, \boldsymbol{\ell}_1 \mid \mathbf{X}_t, \boldsymbol{\ell}_t)$ as a Laplace distribution, whose mean

---

[1]We consider probability paths conditioned on the building blocks $\mathcal{B}$ throughout this section, and omit it from the notation.

$\boldsymbol{\mu}_t^\theta$ is parameterized by the neural network as:

$$\hat{\mathbf{X}}_1, \hat{\boldsymbol{\ell}}_1 = \boldsymbol{\mu}_t^\theta(\mathbf{X}_t, \boldsymbol{\ell}_t \mid \mathcal{B}), \quad (3)$$

where $\hat{\mathbf{X}}_1, \hat{\boldsymbol{\ell}}_1$ are predicted clean Cartesian coordinates and lattice parameters. Then Equation (2) simplifies to minimizing the $L^1$ reconstruction error (Zaghen et al., 2025):

$$\mathcal{L}(\theta) = \mathbb{E}_{\mathcal{S}_1, \mathcal{S}_t, t}\Big[\lambda_{\text{coord}}\|\hat{\mathbf{X}}_1 - \mathbf{X}_1\|_1 + \lambda_{\text{lattice}}\|\hat{\boldsymbol{\ell}}_1 - \boldsymbol{\ell}_1\|_1\Big],$$
$$(4)$$

where $\mathcal{S}_1 = (\mathbf{A}, \mathbf{X}_1, \boldsymbol{\ell}_1) \sim q(\mathcal{S})$, $\mathcal{S}_t = (\boldsymbol{A}, \boldsymbol{X}_t, \boldsymbol{\ell}_t)$, $t \sim \mathcal{U}[0, 1]$, and $\lambda_{\text{coord}}, \lambda_{\text{lattice}}$ are loss weights.

**Prior distribution.** We define the prior distributions over atomic coordinates $\mathbf{X}$ and lattice parameters $\boldsymbol{\ell} = (\boldsymbol{d}, \boldsymbol{\phi})$. For the coordinates, we employ a centered Gaussian prior $\mathbf{X}_0 = \mathbf{P}\varepsilon$, where $\varepsilon \sim \mathcal{N}(0, \mathbf{I})$ and $\mathbf{P} = \mathbf{I} - \frac{1}{N}\mathbf{1}\mathbf{1}^\top$ ensures a zero center of mass. For the lattice parameters, the lengths $\boldsymbol{d}$ are sampled independently from log-normal distributions as $\boldsymbol{d} \sim \text{LogNormal}(\boldsymbol{\mu}, \boldsymbol{\sigma})$, where parameters $\boldsymbol{\mu}$ and $\boldsymbol{\sigma}$ are estimated via maximum likelihood estimation, and the lattice angles $\boldsymbol{\phi}$ are sampled uniformly from the range $[60°, 120°]$ (Miller et al., 2024).

### 3.3. Model Architecture

The denoising network, $\boldsymbol{\mu}_t^\theta(\boldsymbol{X}_t, \boldsymbol{\ell}_t|\mathcal{B})$, employs a Diffusion Transformer (DiT) backbone (Peebles & Xie, 2023; Joshi et al., 2025) organized into three stages: an *encoder* that constructs features from building blocks and noisy inputs, a deep *trunk* for modeling interactions, and a *decoder* for prediction. Detailed pseudocode is provided in Section A.

**Encoder.** The encoder initializes single-atom ($\boldsymbol{Q} \in \mathbb{R}^{N \times D_Q}$) and pairwise ($\boldsymbol{P} \in \mathbb{R}^{N \times N \times D_P}$) representations from the building blocks $\mathcal{B}$. First, we construct initial single features $\boldsymbol{C}$ using atom types, formal charges, and local reference coordinates $\boldsymbol{X}_{\text{local}}$ derived from RDKit (Landrum et al., 2006). Simultaneously, we explicitly featurize the internal geometry of the building blocks to initialize $\boldsymbol{P}$. We compute local displacement vectors $\boldsymbol{D}_{ij} = \boldsymbol{X}_{\text{local},i} - \boldsymbol{X}_{\text{local},j}$ and Euclidean distances $\|\boldsymbol{D}_{ij}\|$, embedding these alongside bond types and block adjacency masks that indicate whether a pair of atoms belong to the same building blocks. We then condition $\boldsymbol{P}$ on $\boldsymbol{C}$ via an MLP, while $\boldsymbol{Q}$ is formed by adding projections of the noisy global coordinates $\boldsymbol{X}_t$ and lattice $\boldsymbol{\ell}_t$ to $\boldsymbol{C}$. These features are processed by a shallow DiT$_{\text{encoder}}$ where $\boldsymbol{P}$ acts as an additive bias to the attention logits (Algorithm 6).

**Trunk.** The trunk performs the main atomic-interaction computation via a deep stack of DiT blocks. We first project the encoder outputs to latent spaces $\boldsymbol{S}$ and $\boldsymbol{Z}$. Crucially, before the transformer pass, we infuse single-atom context into the pairwise bias $\boldsymbol{Z}$: $\boldsymbol{Z}_{ij} \leftarrow \boldsymbol{Z}_{ij} + \text{Linear}(\boldsymbol{S}_i) + \text{Linear}(\boldsymbol{S}_j)$. This updated $\boldsymbol{Z}$ serves as the

structural attention bias (as defined in the encoder) throughout the trunk. Following the trunk computation, we apply a global residual connection to update the encoder features:

$$Q \leftarrow Q + \text{Linear}(S').$$

**Decoder.** The decoder maps the updated features $Q$ to the denoised state. Following a $\text{DiT}_{\text{decoder}}$, we use two parallel heads, consisting of a LayerNorm and a linear layer, to generate the output. The coordinate head projects per-atom features to $\hat{X}_1 \in \mathbb{R}^{N \times 3}$, while the lattice head applies global average pooling followed by a projection to $\hat{\ell}_1 \in \mathbb{R}^6$.

### 3.4. Feynman-Kac Steering with MLIP

To improve the sampling quality, we propose using a machine learning interatomic potential (MLIP) with Feynman-Kac steering (Singhal et al., 2025). Specifically, we sample from an exponentially tilted distribution derived from the base generative model

$$p_{\text{target}}(X, \ell \mid \mathcal{B}) \propto p_\theta(X, \ell \mid \mathcal{B}) \exp\left(-\lambda E(\mathcal{S})\right), \quad (5)$$

where $E(\cdot)$ is the energy estimated by MLIP and $\lambda$ controls the steering strength (see Section B full algorithm).

To perform particle resampling for Feynman-Kac steering, we convert the ODE defined by the learned velocity field $\mathbf{v}(X_t, t)$ into the SDE

$$\mathrm{d}X_t = \left(\mathbf{v}(X_t, t) + \tfrac{1}{2}g(t)^2 s(X_t, t)\right)\mathrm{d}t + g(t)\,\mathrm{d}W_t, \quad (6)$$

where $W_t$ is a standard Wiener process, $g(t)$ is the diffusion coefficient (Ma et al., 2024), and $s(X_t, t)$ is the score function. For a centered Gaussian prior $X_0 = P\varepsilon$ with $\varepsilon \sim \mathcal{N}(0, I)$, the score can be computed directly from $\mathbf{v}$:

$$s(X_t, t) = \frac{t\,\mathbf{v}(X_t, t) - X_t}{1 - t}. \quad (7)$$

## 4. MOF-Only Structure Prediction

### 4.1. Experimental Setup

**Data preprocessing.** We apply a unified preprocessing pipeline to both datasets to decompose MOF structures and extract chemical features. First, we utilize MOFid (Bucior et al., 2019) to separate each MOF into nodes, linkers, solvents, and adsorbates. We validate organic linkers using RDKit and adjust atomic coordinates to ensure the spatial contiguity of each building block. Next, we construct node templates by aggregating local structures from the training set. The BW dataset contains only 9 unique node structures, which we align all instances using the Kabsch algorithm to generate a single average geometry for each type.

**Training and hyperparameters.** We train all variants on 8 NVIDIA B200 GPUs using a three-stage curriculum that progressively increases system size. In the first two stages, we restrict the training subset by a maximum atom count, while the final fine-tuning stage uses the full dataset. We optimize with AdamW (learning rate $10^{-4}$, weight decay 0) and a linear warmup over the first 10k steps. We train three model sizes: ATOMMOF-S (38.4M parameters), ATOMMOF-M (129M), and ATOMMOF-L (491M), for 500, 100, and 500 epochs, respectively. Full architectural and optimization settings are provided in Section C.

### 4.2. Match Rate & RMSD

**Overview.** We evaluate our model on the BW dataset, focusing on the prediction of MOF structures solely from the 2D molecular graphs of their building blocks $\mathcal{B}$. We adopt the 8:1:1 data split introduced by Kim et al. (2025a) for data preprocessing. At inference time, we generate samples using an ODE solver with 10 integration steps. We evaluate performance with 1 and 5 samples, reporting metrics for the best-matching candidate.

**Metrics.** We assess performance using match rate (MR) and root mean squared deviation (RMSD), calculated via pymatgen's StructureMatcher. The matcher compares structures based on specified tolerances for site positions (stol), fractional lengths (ltol), and angles (angle_tol), returning 1 if the structures match within these limits and 0 otherwise. RMSD is computed only for successfully matched pairs. We adopt stol of 0.5 (a typical value for CSP tasks) and 0.3 (the stricter default value), ltol of 0.3, and angle_tol of 10.

**Baselines.** We compare our method against both optimization and learning-based baselines. Optimization-based baselines include random search (RS) and evolutionary algorithms (EA), which iteratively propose candidates until an energetically favorable structure is found (implemented with CrySPY). We also include DiffCSP, a general CSP baseline for inorganic materials, and MOF-specific baselines, i.e., MOFFlow, MOFFlow-2, and MOF-BFN.

Notably, the original MOFFlow and MOF-BFN implementations assume access to ground-truth local geometries, which is unrealistic in practice. To enable a fair comparison, we retrain MOFFlow and MOF-BFN similar to Kim et al. (2025a). Specifically, during training, we use *matched* coordinates, aligning library nodes and RDKit linkers to the ground truth via optimizing rototranslations and torsion angles; during inference, however, we rely on direct predictions of RDKit.

**Results.** As shown in Table 1, both our M and L variants consistently outperform all baselines across every metric. In particular, we observe a large improvement in the stricter evaluation setting (stol = 0.3) and in RMSD; relative to MOFFlow-2, ATOMMOF-L improves the match rate by 35.00% and RMSD by 32.68%. We attribute these gains

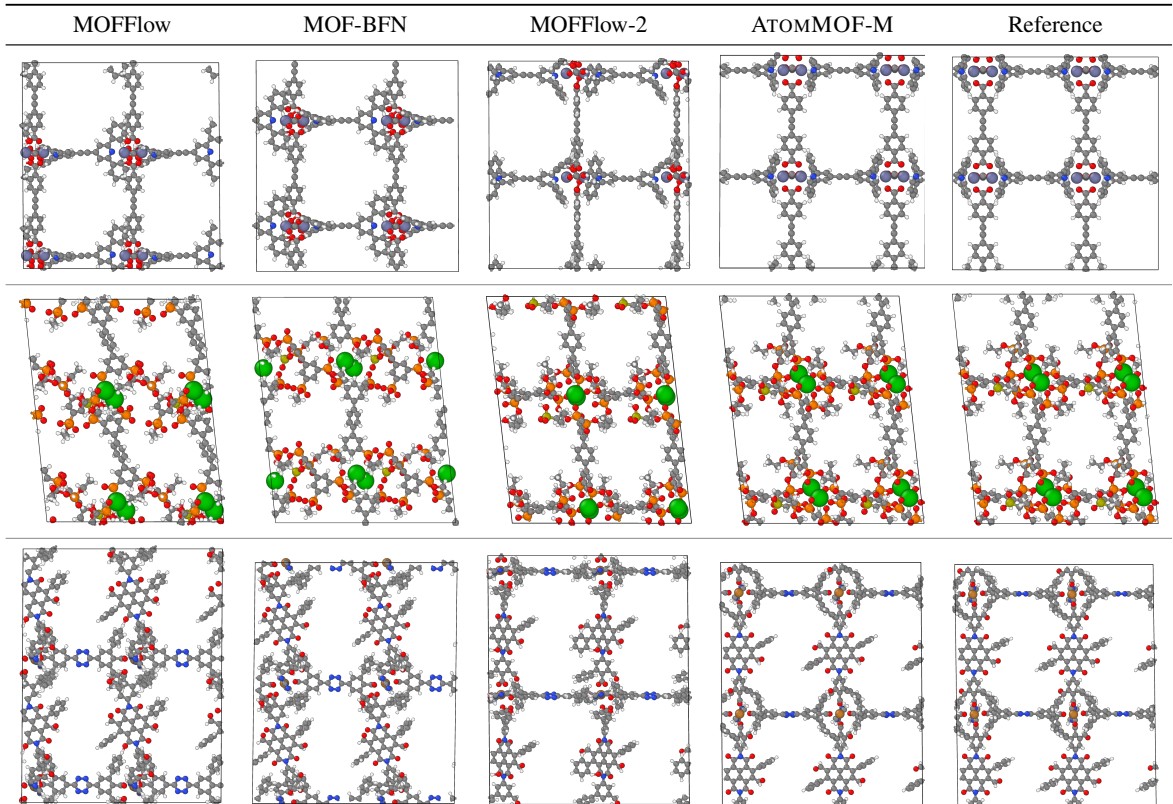

|  | MOFFlow | MOF-BFN | MOFFlow-2 | ATOMMOF-M | Reference |

*Figure 2.* **Qualitative comparison of predicted structures.** Each structure is shown as a $2 \times 2 \times 2$ supercell for visual clarity. Compared to the baselines, ATOMMOF-M more closely matches the ground truth and better avoids steric clashes near the metal nodes, illustrating the advantage of all-atom modeling.

to ATOMMOF's all-atom modeling, whereas prior models assume fixed bond lengths and angles. This advantage is evident in Figure 2, where baselines exhibit steric clashes around metal nodes, while ATOMMOF faithfully reconstructs ground-truth structures.

### 4.3. Property Evaluation

**Overview.** We evaluate whether our model preserves key structural properties relevant to downstream tasks such as gas storage and catalysis. For each generated structure, we measure volumetric surface area (VSA; $\mathrm{m}^2/\mathrm{cm}^3$), gravimetric surface area (GSA; $\mathrm{m}^2/\mathrm{g}$), largest cavity diameter (LCD; Å), pore limiting diameter (PLD; Å), accessible volume (AV; $\text{Å}^3$), unit-cell volume (UCV; $\text{Å}^3$), density ($\mathrm{g}/\mathrm{cm}^3$), and void fraction.

**Metrics & baselines.** We compare against MOFFlow-2 using its default sampling configuration (50 ODE steps), while we find our model outputs sufficiently high-quality output even when using a fewer number of 10 ODE integration steps. All properties are computed with Zeo++ (Willems et al., 2012). We report the RMSD between predicted and ground-truth properties.

**Results.** As shown in Table 2, ATOMMOF achieves lower property errors than MOFFlow-2, and the large variant performs best overall. This suggests that all-atom modeling preserves geometric fidelity better than torsion-based parameterizations with fixed bond constraints, and that increased model capacity further improves property preservation.

### 4.4. Scaling Law

**Parameter scaling.** Figure 3 shows consistent performance improvements as we scale the model from 38.4M (ATOMMOF-S) to 491M (ATOMMOF-L) parameters. Both match rate and RMSD exhibit a strong log-linear trend (Pearson $\pm 0.93$, Spearman $\pm 1.00$), suggesting that scaling laws hold for MOF structure prediction. All results are obtained with 10 integration steps and `stol=0.3`; see Section D for additional results with `stol=0.5`. Qualitative improvements with scale are shown in Figure 4.

**Runtime scaling.** We study how the number of sampling steps affects performance (Figure 5). To this end, we vary the number of ODE integration steps from 1 to 100 and evaluate match rate (`stol=0.3`) on 1,000 randomly sampled test structures. In terms of sampling steps, ATOMMOF-L

*Table 1.* **Comparison of structure prediction accuracy.** MR is the match rate, and RMSD is the root mean squared deviation, and "–" indicates that no structure is matched. `stol` is the site-tolerance, where smaller values impose stricter matching. **Bold** indicates the best result and underlined indicates the second-best. ATOMMOF achieves best performance, with larger gains under stricter tolerance, highlighting the importance of all-atom modeling.

| Model | # Samples | stol = 0.3 | | stol = 0.5 | |
| --- | --- | --- | --- | --- | --- |
| | | MR (%) | RMSD | MR (%) | RMSD |
| RS | 20 | 0.00 | – | 0.00 | – |
| EA | 20 | 0.00 | – | 0.00 | – |
| DiffCSP | 1 | 0.01 | 0.1554 | 0.23 | 0.3896 |
| | 5 | 0.08 | 0.1299 | 0.87 | 0.3982 |
| MOFFlow | 1 | 5.28 | 0.2036 | 21.93 | 0.3329 |
| | 5 | 8.68 | 0.2039 | 32.71 | 0.3290 |
| MOF-BFN | 1 | 6.96 | 0.1979 | 27.66 | 0.3234 |
| | 5 | 12.11 | 0.1950 | 40.15 | 0.3131 |
| MOFFlow-2 | 1 | 8.20 | 0.1894 | 28.71 | 0.3094 |
| | 5 | 15.98 | 0.1842 | 43.95 | 0.2925 |
| ATOMMOF-M | 1 | 10.05 | 0.1328 | 28.63 | 0.2628 |
| | 5 | 19.85 | 0.1249 | 46.08 | 0.2396 |
| ATOMMOF-L | 1 | **11.07** | **0.1275** | **29.96** | **0.2558** |
| | 5 | **21.17** | **0.1199** | **47.47** | **0.2319** |

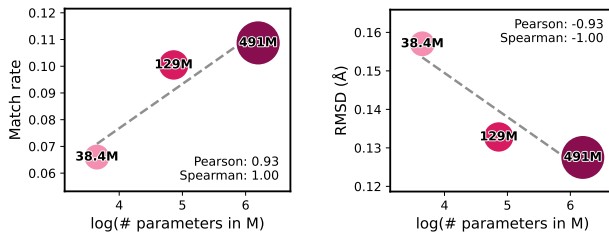

*Figure 3.* **Scaling up ATOMMOF improves performance.** Both match rate (*left*) and RMSD (*right*) exhibit a log-linear relationship with model size, indicating predictable performance gains with increasing number of parameters.

outperforms ATOMMOF-M at every step count, indicating higher sample efficiency per integration step.

In terms of wall-clock time, ATOMMOF-M is stronger at the smallest time budgets, while ATOMMOF-L becomes superior once the runtime exceeds $0.0825$ s. Both variants are substantially more efficient than MOFFlow-2: ATOMMOF surpasses MOFFlow-2's peak match rate ($\sim 7.6\%$) with only 3 steps, and largely saturates by $\sim 10$ steps at $\sim 11.7\%$ (ATOMMOF-L) and $\sim 9.8\%$ (ATOMMOF-M). Results with `stol=0.5` are reported in Section D.

### 4.5. Feynman-Kac Steering with MLIP

**Feynman-Kac steering improves stability and validity.** We apply Feynman-Kac steering to our generation process, using `eSEN-ODAC25-Full` (Sriram et al., 2025) as MLIP guidance with 16 particles, $\lambda = 2.0$, resampling interval of

*Table 2.* **Property evaluation.** RMSE between ground-truth and generated MOFs for key geometric and pore properties computed with Zeo++. ATOMMOF yields lower errors than MOFFlow-2, highlighting the importance of all-atom modeling in preserving structural properties.

| | VSA ↓ | GSA ↓ | AV ↓ | UCV ↓ |
| --- | --- | --- | --- | --- |
| MOFFlow-2 | 312.8 | 443.6 | 485.0 | 427.1 |
| ATOMMOF-M | 226.9 | 282.2 | **210.0** | **153.8** |
| ATOMMOF-L | **215.2** | **275.4** | 241.4 | 192.4 |
| | VF ↓ | PLD ↓ | LCD ↓ | DST ↓ |
| MOFFlow-2 | 0.0445 | 1.3751 | 1.5361 | 0.0312 |
| ATOMMOF-M | 0.0257 | 1.0147 | 0.9616 | 0.0177 |
| ATOMMOF-L | **0.0247** | **1.0131** | **0.9435** | **0.0165** |

| Small | Medium | Large | Reference |
| --- | --- | --- | --- |

*Figure 4.* **Effect of model scaling on predicted structures.** Structural quality improves with model size: larger models better recover the reference topology and reduce steric clashes.

5, start time of $t = 0.8$, and immediate potential (Table 3). As summarized in Table 4, steering reduces the formation energy error (i.e., the difference between the formation energy of the predicted structure and the ground-truth) from $0.127$ to $0.020$ eV/Å (an $84.19\%$ decrease; see Figure 11 for a distribution of energies). Steering also increases MOF validity (as measured by `MOFChecker` (Jablonka, 2023)) from $34.16\%$ to $37.87\%$ ($+10.86\%$) and structural validity (based on pairwise interatomic distances to detect atomic overlap) from $81.70\%$ to $91.70\%$ ($+12.24\%$). Overall, our strategy offers a simple refinement that improves both physical stability and geometric validity. See Section E for steering hyperparameters and metric definitions.

**Sensitivity analysis.** Figure 6 reports an ablation study over the steering hyperparameters – i.e., particles, steering strength ($\lambda$), resampling interval, start time, and potential functions (Table 3) – over 1000 random test samples. Increasing the number of particles and steering strength ($\lambda$) consistently improves both stability and validity. We also find that the *immediate* potential mode performs best, consistent with Hartman et al. (2025). Importantly, applying steering later in the denoising trajectory yields higher structural quality. We attribute this to the guiding MLIP being trained on clean MOF structures, which makes its predictions less reliable in the highly noisy early stages of generation. Additional hyperparameter details and tabular results are provided in Section E.

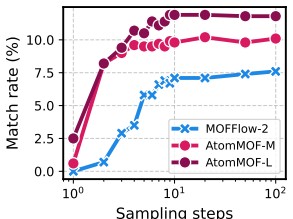 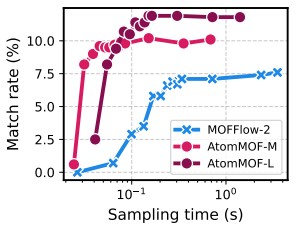

*Figure 5.* **Inference efficiency under scaling.** Match rate (%, stol=0.3) versus (*left*) the number of ODE sampling steps and (*right*) wall-clock sampling time. ATOMMOF-L achieves higher match rates than ATOMMOF-M at the same step or time budget, and both surpass MOFFlow-2 across budgets. In particular, ATOM-MOF exceeds MOFFlow-2's peak match rate with only a few steps and largely saturates within $\sim$10 steps.

*Table 3.* **Overview of potential functions.** Here, $\mathcal{E}_t := E(\hat{\mathcal{S}}_1)$ denotes the potential energy of the clean state predicted at time $t$. $\Delta t$ refers to the resampling interval and $N_t$ denotes the number of steps elapsed up to time $t$.

| Potential | Domain | Definition $G_t(\cdot)$ |
|---|---|---|
| Difference | $\mathcal{S}_{t-\Delta t}, \mathcal{S}_t$ | $\exp\left(\lambda(\mathcal{E}_{t-\Delta t} - \mathcal{E}_t)\right)$ |
| Immediate | $\mathcal{S}_t$ | $\exp\left(-\lambda \mathcal{E}_t\right)$ |
| Max | $\{\mathcal{S}_\tau\}_{\tau \in [0,t]}$ | $\exp\left(-\lambda \min_{\tau \leq t} \mathcal{E}_\tau\right)$ |
| Sum | $\{\mathcal{S}_\tau\}_{\tau \in [0,t]}$ | $\exp\left(-\frac{\lambda}{N_t} \sum_{\tau \leq t} \mathcal{E}_\tau\right)$ |

## 5. MOF-adsorbate Structure Prediction

In this section, we address two questions. First, under a fixed computational budget, how well does our model recover adsorption configurations observed in the dataset (Section 5.1)? Second, when combined with an MLIP during relaxation, can our model identify lower-energy adsorption configurations (Section 5.2)? We use ATOMMOF-L throughout, as it achieves the best performance in Section 4.

**Data preparation.** In general, we follow the setting described in Section 4. We use the filtered ODAC25 training and validation splits. We construct the learning set by extracting local-minimum configurations from each trajectory and applying the preprocessing pipeline in Section 4. This yields 153,249 training datapoints and 3,582 datapoints from the ODAC25 validation split. Since the official test set is not available, we treat the ODAC25 validation split as our test set, and additionally subsample 3,130 datapoints from the training split as a held-out validation set.

To obtain reference coordinates for the metal ions, since the ODAC25 dataset exhibits significantly higher structural diversity with 887 unique nodes, we index ODAC25 nodes by their SMILES strings and randomly sample a specific structure during inference.

### 5.1. Coverage of Adsorption Configurations

**Baselines.** We compare ATOMMOF (10 ODE steps) against grand canonical Monte Carlo (GCMC) and random sam-

*Table 4.* **Effect of Feynman–Kac steering on stability and validity.** MLIP-guided steering improves MOFChecker validity (%, *MOF val.*) and structural validity (%, *Struct. val.*), while substantially reducing the formation energy error (eV/atom, $\Delta E_f$) compared to the unsteered baseline. The last row reports the relative change (%) with respect to no steering.

| Method | MOF val. | Struct. val. | $\Delta E_f$ |
|---|---|---|---|
| No steering | 34.16 | 81.70 | 0.127 |
| Steering | 37.87 | 91.70 | 0.020 |
| Change (%) | +10.86 | +12.24 | −84.19 |

pling (RS). Because these baselines do not generate MOF frameworks, we fix the framework to the structure predicted by ATOMMOF and evaluate only adsorbate placement. We run GCMC in RASPA3 with classical force fields (Ran et al., 2024). For RS, we repeatedly insert a fixed number of adsorbates and reject configurations with steric clashes, defined as any pairwise distance $d_{ij} \leq 0.8(r_i + r_j)$ using covalent radii $r_i, r_j$. See Section F for details.

**Metrics.** We report *coverage* (COV) (Wang et al., 2024) of MOF-adsorbate configurations. Let $\mathcal{D} = \{\mathcal{S}^{(n)}\}_{n=1}^N$ be the set of ground-truth configurations and $\hat{\mathcal{D}}$ the set of predictions; further, let $\hat{\mathcal{D}}_n \subseteq \hat{\mathcal{D}}$ denote the subset of predictions matching $\mathcal{S}^{(n)}$ via StructureMatcher with stol=0.5. We compute the fraction of recovered structures as $\text{COV} = \frac{1}{N} \sum_n \mathbb{I}[|\hat{\mathcal{D}}_n| > 0]$.

**Results.** Figure 7 shows that ATOMMOF achieves higher coverage than both baselines under the same time budget, and is orders of magnitude faster than GCMC. ATOMMOF reaches 23.11% coverage in under 1 s, whereas GCMC requires nearly $10^4$ s to reach 21.06%. RS runs on a time scale comparable to ATOMMOF but consistently attains lower coverage, indicating that ATOMMOF learns a distribution that better matches the reference adsorption configurations.

### 5.2. Adsorption Energy Success Rate

**FK steering details.** We generate 1 and 10 candidates per test datapoint. To ensure stability during MLIP relaxation, we apply Feynman-Kac steering with penalty $E_{\text{overlap}} = \sum_{i<j} \max(0, 0.8(r_i + r_j) - d_{ij})^2$, where $d_{ij}$ is the interatomic distance and $r_i$ is the covalent radius. We filter any remaining overlapping structures via $(d_{ij} \leq 0.8(r_i + r_j))$.

**Metrics.** Similar to Lan et al. (2023); Kolluru & Kitchin (2024), we report success rate (SR). A prediction is a success if its MLIP relaxation converges within 300 steps to a maximum force below 0.05 eV/Å and $\varepsilon_{\min} = \min_{\hat{\mathcal{S}} \in \hat{\mathcal{D}}} \Delta E_{\text{ads}}(\hat{\mathcal{S}}) - \min_{\mathcal{S} \in \mathcal{D}} \Delta E_{\text{ads}}(\mathcal{S}) \leq 0.1$ eV. We compute the adsorption energy with eSEN-ODAC25-Full.

**Results.** Table 5 shows that ATOMMOF achieves success rates of 12.30% with 1 sample and 18.59% with 10 samples. In addition, the model finds configurations with lower

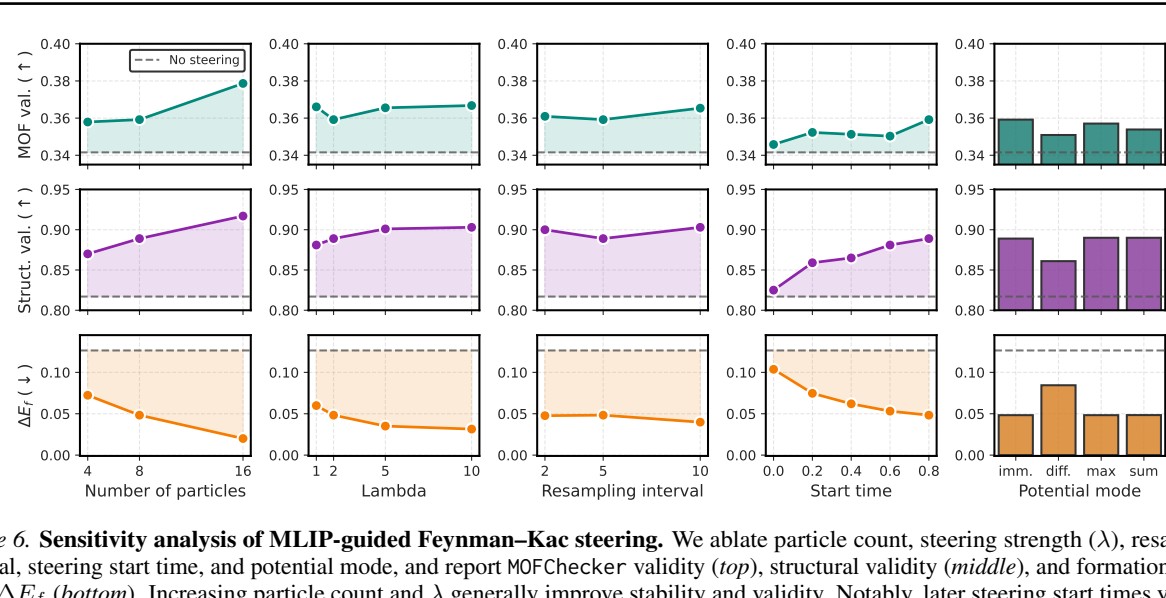

*Figure 6.* **Sensitivity analysis of MLIP-guided Feynman–Kac steering.** We ablate particle count, steering strength ($\lambda$), resampling interval, steering start time, and potential mode, and report MOFChecker validity (*top*), structural validity (*middle*), and formation energy error $\Delta E_f$ (*bottom*). Increasing particle count and $\lambda$ generally improve stability and validity. Notably, later steering start times yield the best performance, leveraging higher MLIP accuracy as structures denoise.

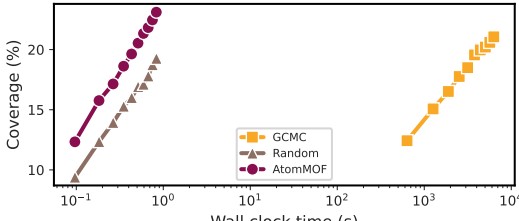

*Figure 7.* **Coverage vs. wall-clock time.** Coverage (%) of reference adsorption configurations versus wall-clock time (log scale) for ATOMMOF, random sampling (RS), and GCMC. ATOMMOF attains the highest coverage at the lowest runtime.

*Table 5.* **Adsorption energy success rate.** Success rate and minimum energy difference ($\varepsilon_{\min}$) for 1 vs. 10 generated samples. Average $\varepsilon_{\min}$ is computed over successfully converged structures. ATOMMOF can find configurations with lower adsorption energies than the reference.

| Metric | 1 Sample | 10 Samples |
|---|---|---|
| Success rate (%) | 12.30 | 18.59 |
| $\varepsilon_{\min} < 0$ (%) | 6.30 | 14.20 |
| Average $\varepsilon_{\min}$ (eV) | 0.0841 | -0.0462 |

adsorption energy than the reference ($\varepsilon_{\min} < 0$) in 6.30% (1 sample) and 14.20% (10 samples) of systems. Among the cases where relaxation converges, the average of the minimum energy differences for 10 samples is $-0.0462$ eV, indicating that the best generated candidates are, on average, slightly lower in energy than the reference minimum. This trend is consistent with Figure 8, which shows that the distribution of minimum adsorption energies from our samples closely matches the reference.

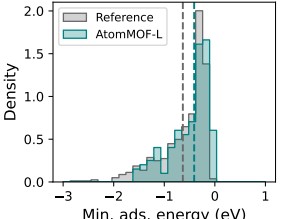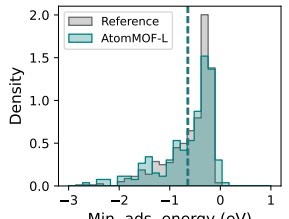

*Figure 8.* **Adsorption energy histogram.** Histogram of the minimum adsorption energy across test systems for 1 (*left*) vs. 10 generated samples (*right*). The distribution closely matches the reference. With 10 samples, ATOMMOF-L achieves a slightly lower mean minimum energy, suggesting that it can provide initializations for finding low-energy adsorption configurations.

## 6. Conclusion

In this work, we present ATOMMOF, a scalable flow-based model for MOF-adsorbate structure prediction. Unlike prior approaches that impose strong structural constraints, e.g., rigid-body assumptions, ATOMMOF models all atoms directly, which we find is necessary for high structural accuracy. We also establish the first scaling laws for porous-crystal generation, showing predictable improvements as model size and compute increase. To further improve sampling robustness, we introduce Feynman-Kac steering guided by an MLIP, which increases the stability and validity of generated structures. Empirically, ATOMMOF outperforms prior baselines on BW. On ODAC25, ATOMMOF recovers diverse adsorption configurations with substantially less wall-clock time than conventional workflows (e.g., GCMC and random sampling), and provides initializations for MLIP relaxation that potentially reach adsorption energies lower than the reference.

## Impact Statement

This paper introduces a scalable flow-matching model for all-atom MOF–adsorbate structure prediction. By accelerating structure prediction and sampling, it may support materials discovery for energy and environmental applications (e.g., gas separations and carbon capture). As with other generative tools, it could be misused to facilitate harmful chemical design. We encourage responsible use with appropriate oversight and compliance with applicable laws and regulations.

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

## A. Model Architecture

Here we describe the detailed implementation of our denoiser, $\boldsymbol{\mu}_t^\theta(\mathbf{X}_t, \boldsymbol{\ell}_t \mid \mathcal{B})$ (Algorithm 1). The architecture consists of three stages: (i) an *encoder* that featurizes the building blocks $\mathcal{B}$ into single-atom and pairwise representations (Algorithm 2); (ii) a *trunk* that performs the main computation over atomic interactions (Algorithm 3); and (iii) a *decoder* that maps the transformed features back to per-atom outputs to predict $\hat{\mathbf{X}}_1$ and $\hat{\boldsymbol{\ell}}_1$ (Algorithm 4).

---

**Algorithm 1** Denoiser $\boldsymbol{\mu}_t^\theta(\mathbf{X}_t, \boldsymbol{\ell}_t \mid \mathcal{B})$

---

    **Input:** Noisy inputs $\boldsymbol{X}_t, \boldsymbol{\ell}_t$, Time $t$, building block $\mathcal{B}$.
    **Output:** Denoised updates $\hat{\mathbf{X}}_1, \hat{\boldsymbol{\ell}}_1$.
    *# 1. Encoding*
    $\boldsymbol{Q}, \boldsymbol{P} \leftarrow \text{Encoder}(\boldsymbol{X}_t, \boldsymbol{\ell}_t, t, \mathcal{B})$
    *# 2. Projection*
    $\boldsymbol{S} \leftarrow \text{Linear}(\boldsymbol{Q})$
    $\boldsymbol{Z} \leftarrow \text{Linear}(\boldsymbol{P})$
    *# 3. Trunk*
    $\boldsymbol{S}' \leftarrow \text{Trunk}(\boldsymbol{S}, \boldsymbol{Z}, t)$
    *# 4. Residual projection*
    $\boldsymbol{Q} \leftarrow \boldsymbol{Q} + \text{Linear}(\boldsymbol{S}')$
    *# 5. Decoding*
    $\hat{\mathbf{X}}_1, \hat{\boldsymbol{\ell}}_1 \leftarrow \text{Decoder}(\boldsymbol{Q}, t)$
    **Return:** $\hat{\mathbf{X}}_1, \hat{\boldsymbol{\ell}}_1$

---

**Algorithm 2** Encoder

---

    **Input:** Noisy coordinates $\boldsymbol{X}_t$, noisy lattice $\boldsymbol{\ell}_t$, time $t$, building block $\mathcal{B}$.
    **Output:** Single representation $\boldsymbol{Q}$, pair representation $\boldsymbol{P}$.
    *# 1. Feature embedding*
    Build $\boldsymbol{f}_{\text{atom}}$ from $\mathcal{B}$ by concatenating RDKit-initialized coordinates $\boldsymbol{X}_{\text{local}}$, formal charges, and atom types.
    Initialize single representation: $\boldsymbol{C} \leftarrow \text{LinearNoBias}(\boldsymbol{f}_{\text{atom}})$
    *# 2. Pairwise representation construction*
    Compute pairwise distances $\boldsymbol{D}_{ij} \leftarrow \boldsymbol{X}_{\text{local},i} - \boldsymbol{X}_{\text{local},j}$ and norms $\|\boldsymbol{D}_{ij}\|$.
    Construct block adjacency mask $\boldsymbol{v}_{ij}$.
    Initialize pair representation $\boldsymbol{P}$:
    $\boldsymbol{P} \leftarrow \text{Embed}_{\text{pos}}(\boldsymbol{D}) + \text{Embed}_{\text{dist}}(\|\boldsymbol{D}\|) + \text{Embed}_{\text{bond}}(\mathcal{B}_{\text{bond}})$.
    $\boldsymbol{P} \leftarrow \boldsymbol{P} \odot \boldsymbol{v} + \text{Embed}_{\text{mask}}(\boldsymbol{v}) \odot \boldsymbol{v}$.
    *# 3. Condition integration*
    Add single context to pairs:
    $\boldsymbol{P}_{ij} \leftarrow \boldsymbol{P}_{ij} + \text{MLP}_Q(\boldsymbol{C}_i) + \text{MLP}_K(\boldsymbol{C}_j) + \text{MLP}_{\text{pair}}(\boldsymbol{P}_{ij})$.
    Add noise conditioning to single rep $\boldsymbol{Q}$:
    $\boldsymbol{Q} \leftarrow \boldsymbol{C} + \text{Linear}(\boldsymbol{X}_t) + \text{Linear}(\boldsymbol{\ell}_t)$.
    *# 4. Encoding*
    $\boldsymbol{Q}, \boldsymbol{P} \leftarrow \text{DiT}_{\text{encoder}}(\boldsymbol{Q}, \boldsymbol{P}, t)$.
    **Return:** $\boldsymbol{Q}, \boldsymbol{P}$

---

---

**Algorithm 3** Trunk

---

**Input:** Single representation $S_{\text{in}}$, pair representation $Z_{\text{in}}$, time $t$.
**Output:** Updated single representation $S'$.
*# 1. Initialization*
$S \leftarrow \text{Linear}_{\text{init}}(S)$
$Z_{\text{bias}} \leftarrow \text{Linear}_1(S) \oplus \text{Linear}_2(S)^T$
$Z \leftarrow Z + Z_{\text{bias}}$
*# 2. Normalization*
$S \leftarrow \text{Linear}(\text{LayerNorm}(S))$, $Z \leftarrow \text{Linear}(\text{LayerNorm}(Z))$
*# 3. Transformer Pass*
$S_{\text{out}} \leftarrow \text{DiT}_{\text{trunk}}(S, Z, t)$
**Return:** $S'$

---

---

**Algorithm 4** Decoder

---

**Input:** Single representation $Q$, time $t$.
**Output:** Coordinate prediction $\hat{X}_1$, lattice prediction $\hat{\ell}_1$.
*# 1. Feature decoding*
$Q_{\text{dec}} \leftarrow \text{DiT}_{\text{decoder}}(Q, t)$
*# 2. Coordinate projection*
$\hat{X}_1 \leftarrow \text{Linear}(\text{LayerNorm}(Q_{\text{dec}}))$
$Q_{\text{global}} \leftarrow \text{Average}(Q_{\text{dec}})$
$\hat{\ell}_1 \leftarrow \text{Linear}(\text{LayerNorm}(Q_{\text{global}}))$
**Return:** $\hat{X}_1, \hat{\ell}_1$

---

---

**Algorithm 5** Diffusion Transformer (DiT) Architecture

---

**Input:** Input $S$, time $t$, (optional) bias $Z$.
**Output:** Updated $S$.
*# 1. Time conditioning*
$c \leftarrow \text{MLP}(\text{SinusoidalEmbedding}(t))$
**for** $l = 1 \ldots L$ **do**
  *# 2. Adaptive modulation (AdaLN)*
  $\gamma_{1,2}, \beta_{1,2}, \alpha_{1,2} \leftarrow \text{Chunk}(\text{Linear}(\text{SiLU}(c)), 6)$
  *# 3. Self-attention block*
  $S_{\text{norm}} \leftarrow \text{LayerNorm}(S) \cdot (1 + \gamma_1) + \beta_1$
  $S \leftarrow S + \alpha_1 \cdot \text{Attn}(S_{\text{norm}}, \text{bias} = Z)\text{s}$
  *# 4. Feed-forward block*
  $S_{\text{norm}} \leftarrow \text{LayerNorm}(S) \cdot (1 + \gamma_2) + \beta_2$
  $S \leftarrow S + \alpha_2 \cdot \text{MLP}(S_{\text{norm}})$
**end for**
**Return:** $S$

---

---

**Algorithm 6** Attention with pairwise bias

---

**Input:** Input $X$, (optional) pairwise bias $Z$.
**Output:** Attention output $Y$
$\mathsf{Q}, \mathsf{K}, \mathsf{V} \leftarrow \text{Linear}_{q,k,v}(X)$
$A \leftarrow \text{softmax}\left(\frac{\mathsf{Q}\mathsf{K}^{\top}}{\sqrt{D_k}} + \text{Linear}_{\text{bias}}(Z)\right)\mathsf{V}$
$Y \leftarrow \text{Linear}_{\text{out}}(A)$
**Return:** $Y$

---

# B. Feynman-Kac Steering Algorithm

In this section, we present the Feynman–Kac (FK) steering algorithm guided by an MLIP model. The key idea is to bias the sampling trajectory towards low-energy structures by maintaining an ensemble of $K$ particles, updating their weights at intermediate steps using the potential, and resampling the particles according to these weights.

---

**Algorithm 7** Feynman–Kac Steering with MLIP

---

**Input:** Denoiser $\boldsymbol{\mu}_\theta$, MLIP $E(\cdot)$.

**Hyperparams:** Steering strength $\lambda$, particles $K$, resampling interval $\Delta\tau$.

**Initialize:** Sample $\mathcal{S}_0^{(k)} \sim p_0$ for $k = 1, \ldots, K$.

**Initialize:** History of energies $\mathcal{E}_{\text{hist}}^{(k)} \leftarrow \emptyset$ for all $k$.

Define time discretization $\{t_i\}_{i=0}^T$ with $t_0 = 0, t_T = 1$.

**for** $i = 0$ **to** $T - 1$ **do**

    $t \leftarrow t_i, \Delta t \leftarrow t_{i+1} - t_i$.

    *# 1. Predict clean state*

    $\hat{\mathcal{S}}_1^{(k)} \leftarrow \boldsymbol{\mu}_t^\theta(\boldsymbol{X}_t^{(k)}, \boldsymbol{\ell}_t^{(k)} \mid \mathcal{B})$ for all $k$.

    Compute current energies: $\mathcal{E}_{\text{curr}}^{(k)} \leftarrow E(\hat{\mathcal{S}}_1^{(k)})$.

    *# 2. Resample (Feynman–Kac)*

    **if** $t \geq t_{\text{start}}$ **and** $i \bmod \Delta\tau = 0$ **then**

        Compute weights $w_k$ using $G_t$ (see Table 3).

        *e.g., for difference:* $w_k \leftarrow \exp(\lambda(\mathcal{E}_{t-\Delta t}^{(k)} - \mathcal{E}_t^{(k)}))$.

        Normalize weights: $\tilde{w}_k \leftarrow w_k / \sum_{j=1}^K w_j$.

        Resample indices $\{j_k\}_{k=1}^K \sim \text{Multinomial}(\{\tilde{w}_k\})$.

        Update particles: $\mathcal{S}_t^{(k)} \leftarrow \mathcal{S}_t^{(j_k)}, \mathcal{E}_{\text{curr}}^{(k)} \leftarrow \mathcal{E}_{\text{curr}}^{(j_k)}$.

    **end if**

    Update history: $\mathcal{E}_{\text{hist}}^{(k)} \leftarrow (\mathcal{E}_{\text{hist}}^{(k)}, \mathcal{E}_{\text{curr}}^{(k)})$

    *# 3. SDE update (for all $k$)*

    *# Step lattice:*

    $\mathbf{v}_{t,\boldsymbol{\ell}}^{(k)} \leftarrow (\hat{\boldsymbol{\ell}}_1^{(k)} - \boldsymbol{\ell}_t^{(k)})/(1-t)$

    $\boldsymbol{\ell}_{t+\Delta t}^{(k)} \leftarrow \boldsymbol{\ell}_t^{(k)} + \mathbf{v}_{t,\boldsymbol{\ell}}^{(k)} \Delta t$

    *# Step coordinates:*

    $\mathbf{v}_{t,\boldsymbol{X}}^{(k)} \leftarrow (\hat{\boldsymbol{X}}_1^{(k)} - \boldsymbol{X}_t^{(k)})/(1-t)$

    $s_t^{(k)} \leftarrow (t\mathbf{v}_{t,\boldsymbol{X}}^{(k)} - \boldsymbol{X}_t^{(k)})/(1-t)$, sample $\boldsymbol{z}^{(k)} \sim \mathcal{N}(\boldsymbol{0}, \boldsymbol{I})$, compute $g(t) \leftarrow \sqrt{(1-t)/2}$

    $\boldsymbol{X}_{t+\Delta t}^{(k)} \leftarrow \boldsymbol{X}_t^{(k)} + (\mathbf{v}_{t,\boldsymbol{X}}^{(k)} + \frac{1}{2}g(t)^2 s_t^{(k)})\Delta t + g(t)\sqrt{\Delta t}\, \boldsymbol{z}^{(k)}$

**end for**

**Return:** $\{\mathcal{S}_1^{(k)}\}_{k=1}^K$

---

## C. Implementation Details

We trained all model variants using 8 NVIDIA B200 GPUs. The specific architectural hyperparameters for ATOMMOF-S, ATOMMOF-M, and ATOMMOF-L are detailed in Table 6. To handle the high variability in atomic counts, we employed dynamic batching where the batch size is determined by a squared-atom limit ($N_{\max}^2$).

Optimization was performed using the AdamW optimizer with a linear warmup scheduler. The specific loss weights and optimizer settings are listed in Table 7.

To improve convergence and stability, we adopt a multi-stage curriculum learning strategy, where we progressively increase the complexity of the training data by increasing the maximum number of atoms used in each stage (Table 8).

*Table 6.* Hyperparameters for ATOMMOF-S, ATOMMOF-M, and ATOMMOF-L.

| Parameter | ATOMMOF-S | ATOMMOF-M | ATOMMOF-L |
|---|---|---|---|
| **Encoder & Decoder** | | | |
| Single dim. | 256 | 256 | 256 |
| Pairwise dim. | 128 | 128 | 128 |
| Depth | 3 | 3 | 3 |
| Heads | 4 | 4 | 4 |
| **Trunk** | | | |
| Single dim. | 256 | 512 | 1024 |
| Pairwise dim. | 128 | 256 | 512 |
| Depth | 24 | 24 | 24 |
| Heads | 8 | 8 | 16 |
| **Total params.** 38.4M | 129M | 491M | |
| **Batch limit** ($N_{max}^2$) | | | |
| Pretraining | $3.0 \times 10^6$ | $1.0 \times 10^6$ | $1.0 \times 10^6$ |
| Fine-tuning 1 | $3.0 \times 10^6$ | $1.0 \times 10^6$ | $1.0 \times 10^6$ |
| Fine-tuning 2 | $2.5 \times 10^6$ | $8.0 \times 10^5$ | $8.0 \times 10^5$ |

*Table 7.* Training and optimization hyperparameters.

| Parameter | Value |
|---|---|
| **Loss weights** | |
| Coordinate weight | 1.0 |
| Lengths weight | 1.0 |
| Angle weight | 0.01 |
| **Optimizer (AdamW)** | |
| Learning rate | $1 \times 10^{-4}$ |
| Weight decay | 0.0 |
| **Scheduler (linear warmup)** | |
| Base LR | 0.0 |
| Warmup steps | 10,000 |

*Table 8.* **Curriculum learning schedule.** Maximum atom count and training epochs for each stage.

| Setting | Pretraining | Fine-tuning 1 | Fine-tuning 2 |
|---|---|---|---|
| **Maximum atom count** | | | |
| BW | 200 | 300 | No limit |
| ODAC25 | 300 | 400 | No limit |
| **Epochs** | 500 | 100 | 500 |

# D. Additional Results for Scaling Law & Inference Efficiency

We here present additional results for the scaling law and inference efficiency under `stol=0.5` condition.

**Scaling law.** Figure 9 demonstrates that the scaling benefits persist under the `stol=0.5` condition. As we scale from 38.4M to 491M parameters, we observe consistent improvements in generation quality. Both match rate and RMSD maintain a strong log-linear trend with model size (Pearson 0.92 and $-0.94$, respectively; Spearman $\pm 1.00$), confirming that the predictable performance gains observed in the main text are robust to changes in the evaluation threshold.

**Inference efficiency.** As shown in Figure 10, the performance disparity between models narrows under this looser matching threshold compared to the stricter `stol=0.3` setting. While ATOMMOF-L remains the top-performing model overall, ATOMMOF-M saturates at a match rate slightly lower than that of MOFFlow-2 given a larger computational budget ($> 10$ steps). However, ATOMMOF retains a critical advantage in sampling efficiency. Both ATOMMOF-M and ATOMMOF-L achieve high match rates ($> 20\%$) in just 2 integration steps, a regime where the performance of MOFFlow-2 is still negligible ($< 10\%$). This confirms that ATOMMOF offers a better trade-off between inference cost and sample quality, providing quality structures with a lower budget than the baseline.

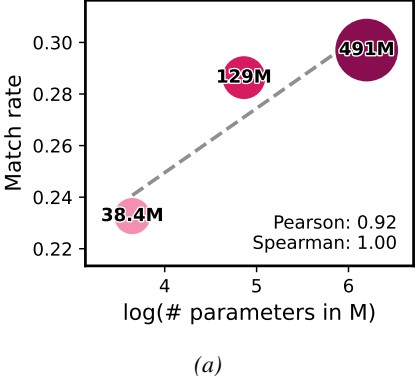
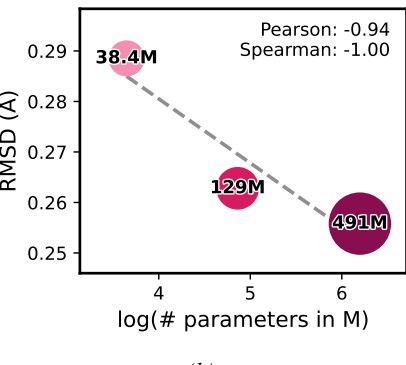

*(a)*        *(b)*

*Figure 9.* **Scaling law analysis (`stol=0.5`).** Consistent with the main text results, scaling up the model size leads to predictable gains even under a looser matching threshold. Both match rate (a) and RMSD (b) exhibit a strong log-linear relationship with the number of parameters.

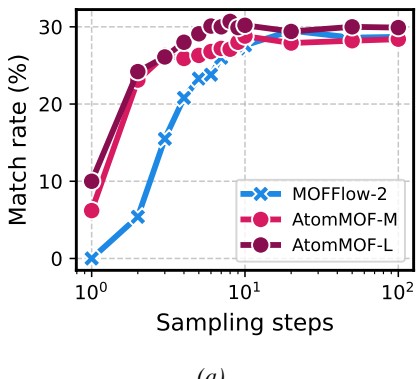
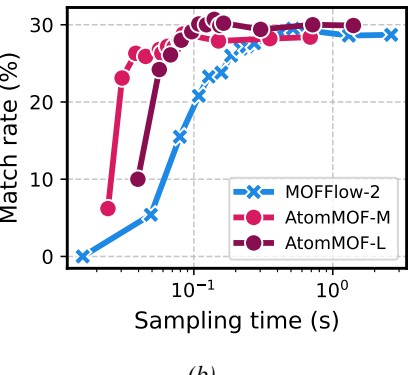

*(a)*        *(b)*

*Figure 10.* **Inference efficiency comparison (`stol=0.5`).** Match rate as a function of (a) ODE sampling steps and (b) wall-clock sampling time. While the performance disparity narrows under looser threshold, ATOMMOF-L remains the top-performing model. Although ATOMMOF-M saturates slightly below MOFFlow-2 at larger computational budgets ($> 10$ steps), ATOMMOF maintains a critical efficiency advantage: both variants achieve $> 20\%$ match rate within just 2 steps (compared to $< 10\%$ for the baseline), offering a superior trade-off between sample quality and inference cost.

## E. Feynman–Kac Steering Experimental Details

Here we provide additional details on the Feynman–Kac steering experiments in Section 4.5. All results use the ATOMMOF-M model and are obtained by integrating the SDE for 200 steps.

### E.1. Metric Definitions

We define three metrics: MOFChecker validity, structure validity, and formation energy error.

1. **MOFChecker validity** (Jablonka, 2023) (%): an MOF is considered valid if it passes all MOFChecker checks, which include (but are not limited to) the presence of key elements (e.g., C/H and a metal), no atomic overlaps, chemically reasonable coordination environments (e.g., around C and N), sufficient porosity, no unphysically large charges, and no isolated molecules.

2. **Structure validity** (%): a predicted crystal structure is considered valid if (i) all pairwise interatomic distances are at least $0.5$ Å, and (ii) the unit-cell volume is at least $0.1$ Å$^3$.

3. **Formation energy error** (eV/atom): the difference between the formation energy of the predicted structure and that of the ground-truth structure, which measures the stability of the predicted structure relative to the ground truth (negative values indicate a more stable prediction):

$$\Delta E_f \ := \ \Delta E_f(\hat{\mathcal{S}}) - \Delta E_f(\mathcal{S}). \tag{8}$$

### E.2. Comparison of Energy Distributions

We compare the energy distributions (computed with eSEN-ODAC25-Full) of ATOMMOF samples with and without MLIP-guided FK steering (Section 4.5). As shown in Figure 11, steering brings the distribution closer to the reference.

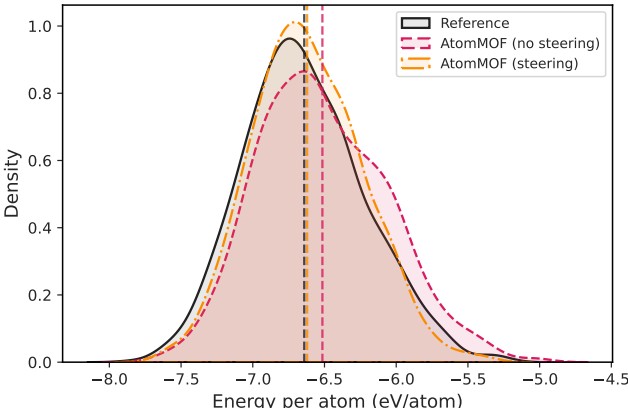

*Figure 11.* **Energy distribution with and without MLIP-guided Feynman–Kac steering.** Kernel density estimates of energies for ATOMMOF samples generated with and without MLIP-guided Feynman–Kac steering, compared to the reference distribution. Steering shifts the distribution toward the reference energy range.

### E.3. Sensitivity Analysis

This section summarizes the hyperparameter settings used in the Feynman–Kac steering sensitivity analysis (Section 4.5). To isolate the effect of each hyperparameter, we fix a default configuration and vary one parameter at a time while keeping all others unchanged. The default settings are listed in Table 9.

For completeness, we report the numerical results underlying Figure 6 in Table 11.

To evaluate the robustness of the method, we conducted ablation studies by varying individual hyperparameters. Table 10 summarizes the specific values tested for each parameter.

*Table 9.* **Default configurations for Feynman-Kac steering sensitivity analysis.** The hyperparameters we consider are the number of particles ($K$), steering strength ($\lambda$), resampling interval ($\Delta\tau$), steering start time $t_{\text{start}}$, and the potential ((Hartman et al., 2025).

| Parameter | Value |
|-----------|-------|
| $K$ | 8 |
| $\lambda$ | 2.0 |
| $\Delta\tau$ | 5 |
| $t_{\text{start}}$ | 0.8 |
| Potential | Immediate |

*Table 10.* **Hyperparameter variations for sensitivity analysis.** For each experiment, one parameter was varied according to the values below, while all others remained at their default values.

| Parameter | Varied Values |
|-----------|---------------|
| $K$ | 4, 16 |
| $\lambda$ | 1.0, 5.0, 10.0 |
| $\Delta\tau$ | 2, 5, 10 |
| $t_{\text{start}}$ | 0, 0.20, 0.40, 0.60, 0.80 |
| Potential | Difference, Max, Sum |

*Table 11.* Numerical results for the FK steering sensitivity analysis (Figure 6).

| Parameter | Value | MOFChecker validity (%) | Structure validity (%) | $\Delta E_f$ (eV/atom) |
|-----------|-------|-------------------------|------------------------|------------------------|
| $K$ | 4 | 35.79 | 87.00 | 0.07235 |
| | 8 | 35.92 | 88.90 | 0.04828 |
| | 16 | 37.87 | 91.70 | 0.02005 |
| $\lambda$ | 1.0 | 36.61 | 88.10 | 0.05987 |
| | 2.0 | 35.92 | 88.90 | 0.04828 |
| | 5.0 | 36.55 | 90.10 | 0.03508 |
| | 10.0 | 36.68 | 90.30 | 0.03148 |
| $\Delta\tau$ | 2 | 36.10 | 90.00 | 0.04767 |
| | 5 | 35.92 | 88.90 | 0.04828 |
| | 10 | 36.54 | 90.30 | 0.03983 |
| $N_{\text{start}}$ | 0.0 | 34.58 | 82.50 | 0.10370 |
| | 0.2 | 35.23 | 85.90 | 0.07469 |
| | 0.4 | 35.13 | 86.50 | 0.06201 |
| | 0.6 | 35.03 | 88.10 | 0.05314 |
| | 0.8 | 35.92 | 88.90 | 0.04828 |
| Potential | Immediate | 35.92 | 88.90 | 0.04828 |
| | Difference | 35.09 | 86.10 | 0.08443 |
| | Max | 35.71 | 89.00 | 0.04828 |
| | Sum | 35.39 | 89.00 | 0.04838 |

## F. Additional Details for MOF–Adsorbate Structure Prediction

Here we provide experimental details for MOF–adsorbate structure prediction experiments in Section 5

**GCMC implementation.** For MOF atoms, we use a mixture of the Dreiding force field (Mayo et al., 1990) and the Universal Force Field (UFF) (Rappé et al., 1992). For adsorbates, we use TraPPE for $CO_2$, $O_2$, and $N_2$ (Potoff & Siepmann, 2001; Zhang & Siepmann, 2006), and TIP4P-ew for $H_2O$ (Horn et al., 2005). Electrostatic interactions use DDEC6 charges (Manz & Sholl, 2010) predicted by PACMAN (Zhao & Chung, 2024) for MOF atoms. Long-range electrostatics are computed with Ewald summation using a force tolerance of $10^{-6}$ kcal/mol/Å, and all pairwise interactions use a 12.8 Å cutoff. Simulations are performed at 298 K.

During Monte Carlo sampling, translation, rotation, and reinsertion moves are attempted with equal probability (1:1:1). For each fixed adsorbate type and loading, we run 10,000 initialization cycles followed by 10,000 production cycles, and take the final production configuration as the representative MOF–adsorbate structure.

**Atomic overlap criterion for RS.** During random sampling (RS), we reject insertions that introduce steric clashes. We compute all framework–adsorbate and inter-adsorbate distances $d_{ij}$ under periodic boundary conditions using the minimum image convention. A configuration is clashed if any atom pair satisfies $d_{ij} \leq 0.8\,(r_i + r_j)$, where $r_i$ and $r_j$ are the covalent radii of atoms $i$ and $j$. Covalent radii are taken from the Atomic Simulation Environment (ASE) (Larsen et al., 2017).

