# OpenReview forum: "AtomMOF: All-Atom Flow Matching for MOF-Adsorbate Structure Prediction"
_ICML.cc/2026/Conference — Submitted to ICML 2026_

### Official Review · Reviewer_9M86 · 2026-03-12

**Soundness:** 4
**Presentation:** 4
**Significance:** 4
**Originality:** 3
**Overall Recommendation:** 5
**Confidence:** 4

**Summary:**

The paper introduces AtomMOF, an all-atom Flow Matching framework for MOF-Adsorbate structure prediction. AtomMOF relaxes the coarse-graining constraints in MOF CSP, and expands to MOF-adsorbate scenarios. The authors also introduce Feynman-Kac steering with MLIP to improve generation stability and validity. Experimental results show the superior performance and the scaling law behavior of AtomMOF.

**Compliance With Llm Reviewing Policy:**

Affirmed.

**Final Justification:**

After reading the rebuttal and the discussion, my major concerns have been sufficiently addressed, and I believe this work makes a valuable contribution to the community.

Given the authors’ commitment to revise the manuscript and include the additional experiments (especially the scaling experiments and the MOF-adsorbate experiments for MOFFlow-2), I will raise my score to 5.

**Key Questions For Authors:**

1. What is the main technical novelty of AtomMOF’s architectural design? Which components are inherited from prior work, and which are newly introduced for MOF-specific modeling?
2. For the success rate metric in the MOF-adsorbate experiments, could AtomMOF be compared against stronger baselines such as AdsorbDiff?
3. AtomMOF adopts a three-stage, AF3-style curriculum learning schedule. To what extent might its advantage over the baselines stem from this different training recipe rather than the model architecture itself?

**Limitations:**

The paper does not explicitly discuss its limitations. Including a discussion of potential limitations would further strengthen the paper.

**Strengths And Weaknesses:**

**Strength**
1. The all-atom nature of AtomMOF allows it to possess full flexibility in MOF generation. The architecture of AtomMOF is neat and the adoption of both single-atom and pairwise representations resembles AF3 [1].
2. AtomMOF shows impressive scalability performance which verifies the potential of its all-atom architecture.
3. The inclusion of MOF-adsorbate structure prediction task is meaningful and relates to the real-world application of direct air capture.

**Weaknesses**
1. The paper lacks a thorough discussion of the principle behind the architectural design of AtomMOF. It would be better if the authors explicitly discuss how AtomMOF relates to other atomic all-atom architectures like ADiT [2], Mofasa [3] and more relevantly AF3. This may undermine the technical novelty of AtomMOF.
2. For the MOF-adsorbate experiments, current evaluation on success rate is based on MLIP. This is different from and less accurate than that in AdsorbDiff [4], which is conducted by DFT.
3. The paper lacks a model parameter comparison with baselines (MOFFlow-2, MOF-BFN, MOFFlow).

[1] Accurate structure prediction of biomolecular interactions with AlphaFold 3, Nature 2024

[2] All-atom Diffusion Transformers: Unified generative modelling of molecules and materials, ICML 2025

[3] Mofasa: A Step Change in Metal-Organic Framework Generation, Arxiv 2025

[4] AdsorbDiff: Adsorbate Placement via Conditional Denoising Diffusion, ICML 2024

---

> ### Author Rebuttal · Authors · 2026-03-31
>
> Dear Reviewer 9M86,
>
> We sincerely thank you for your thorough and constructive review. We are grateful for your positive assessment of our work. Below, we address each concern in detail.
>
> **W1/Q1. Architectural novelty relative to ADiT, Mofasa, and AF3 is insufficiently discussed; unclear which components are inherited versus newly introduced for MOF.**
>
> Indeed, our architecture is largely inspired by AF3, with adaptations to reduce computational cost. Unlike AF3, which updates pairwise representations via the Pairformer $O(N^3d)$, AtomMOF uses attention bias $O(N^2d)$ while retaining pairwise representations, which is important in maintaining performance. This is distinct from ADiT and Mofasa, which use latent diffusion with a VAE and lack pairwise representations.
>
> However, we would like to clarify that our technical contribution is beyond architecture design. We demonstrate, for the first time, scaling laws for porous crystal generation, and introduce MLIP-guided Feynman-Kac steering to improve the structural validity and stability of generated samples. Both contributions are architecture-agnostic and applicable to future models and even across domains.
>
> **W2. ODAC25 evaluation relies on MLIP rather than DFT (unlike AdsorbDiff).**
>
> We agree that DFT evaluation would be ideal. However, DFT on MOFs is substantially more expensive than on the Open Catalyst dataset used by AdsorbDiff; our structures average ~220 atoms per unit cell (up to ~670), compared to ~80 (up to ~220) for OC20-dense used by AdsorbDiff. This translates to roughly 20× higher computational cost per structure, given the O(N³) scaling of DFT. Additionally, AtomMOF predicts the full MOF structure from scratch, whereas AdsorbDiff assumes ground-truth catalyst structures, meaning our predicted structures may require significantly more DFT steps to converge.
>
> To ensure the robustness of our evaluation despite the absence of DFT, we evaluated with UMA (`uma-s-1p2`) as an independent MLIP alongside eSEN:
>
> | Samples | Eval MLIP | Success rate (%) | $\varepsilon_{\text{min}} <0$ (%) | Average $\varepsilon_{\text{min}}$ (eV) |
> | --- | --- | --- | --- | --- |
> | 1 | eSEN | 12.30 | 6.30 | 0.0841 |
> |  | UMA | 13.03 | 6.74 | 0.0455 |
> | 10 | eSEN | 18.59 | 14.20 | -0.0462 |
> |  | UMA | 19.26 | 15.41 | -0.0045 |
>
> The agreement between two independently trained MLIPs  provides stronger confidence in our results. Full DFT validation on a representative subset remains valuable future work.
>
> **W3. Model parameter counts are not compared with baselines (MOFFlow-2, MOF-BFN, MOFFlow).**
>
> We thank the reviewer for this suggestion and report parameter counts below:
>
> | Model | # Params (M) |
> |-------|-------------|
> | MOFFlow | 22.5 |
> | MOF-BFN | 11.2 |
> | MOFFlow-2 | 146 |
> | AtomMOF-M | 129 |
> | AtomMOF-L | 491 |
>
> MOFFlow and MOF-BFN are substantially smaller, but operate under rigid-body assumptions, predicting only rotations and translations ($6M$ degrees of freedom, where $M\approx 7$ building blocks) rather than all-atom coordinates ($3N$, where $N\approx 150$ atoms). This greatly reduces their output dimensionality but limits flexibility and reconstruction accuracy, as shown in our experiments. At comparable parameter count, AtomMOF-M (129M) outperforms MOFFlow-2 (146M) while predicting all-atom coordinates without structural constraints (i.e., fixed bond lengths/angles).
>
> **Q2. For the success rate metric in the MOF-adsorbate experiments, could AtomMOF be compared against stronger baselines such as AdsorbDiff?**
>
> We appreciate the suggestion. However, a direct comparison is not straightforward due to fundamental differences in task formulation: AdsorbDiff assumes a ground-truth catalyst surface and learns only adsorbate placement on 2D surfaces, whereas AtomMOF jointly predicts the full 3D MOF structure and adsorbate configuration from scratch. Additionally, AdsorbDiff was trained on OC20-dense, a different dataset from ODAC25. These differences in both task scope and data make a fair comparison infeasible.
>
> **Q3. Unclear whether improvements stem from the architecture or the AF3-style curriculum training recipe.**
>
> To isolate the contribution of curriculum learning, we re-trained AtomMOF-M without curriculum (training on the full dataset from the start) for a comparable number of training steps (~863K).
>
> | Method | MR (stol=0.3) | RMSD (stol=0.3) | MR (stol=0.5) | RMSD (stol=0.5) |
> | --- | --- | --- | --- | --- |
> | AtomMOF-M without curriculum | 8.93 | 0.1389 | 27.14 | 0.2699 |
> | AtomMOF-M with curriculum | 10.05 | 0.1328 | 28.63 | 0.2628 |
>
> Even without curriculum learning, AtomMOF-M substantially outperforms MOFFlow-2 on RMSD (−26.7% and −12.8% for stol=0.3 and 0.5) and on the stricter MR (stol=0.3), mirroring the same pattern as the curriculum-trained model. This confirms that the architecture is the primary driver of improvement; curriculum learning provides complementary gains on top of an already strong architecture.

---

> > ### Author Rebuttal · Reviewer_9M86 · 2026-04-04
> >
> > Thank the authors for the additional information and for the substantial effort put into the rebuttal. I have the following follow-up comments:
> > 1. **Regarding the architectural novelty**: I understand and acknowledge that the main contribution of the paper is not its architectural novelty. That said, I believe the paper would benefit from a more explicit discussion of how AtomMOF is adapted from, or built upon, prior work. Such a discussion would help readers better position AtomMOF within the broader literature and understand which parts are inherited versus newly introduced for the MOF setting.
> >
> > 2. **Regarding the MOF-adsorbate experiments**: I understand that a direct comparison with AdsorbDiff may not be straightforward, since it would likely require substantial adaptation in both data and modeling assumptions (e.g., 2D surface placement vs. 3D porous environments). However, I still wonder whether it would be possible to include a stronger *learning-based generative* baseline for this setting. The core question here is whether the advantage of AtomMOF in MOF-only structure modeling actually carries over to the MOF-adsorbate setting.
> >
> > 3. **Regarding technical novelty**: In the rebuttal, the authors highlight two main technical contributions: (1) the first demonstration of scaling laws for porous crystal generation, and (2) the introduction of MLIP-guided Feynman-Kac steering. I think both points are interesting, but the paper would benefit from a clearer discussion of their originality and scope. For (1), it is unclear whether the observed scaling behavior is a unique property of AtomMOF or a more general phenomenon that may also hold for other MOF generation methods (e.g., MOFFlow or MOFFlow-2). For (2), Feynman-Kac steering was originally proposed as a general framework for diffusion models, so it would be helpful to clarify what is technically non-trivial in adapting it to the MOF setting with MLIP guidance. This would help readers better understand the paper’s technical originality.
> >
> > 4. **Regarding limitations**: I also believe the paper would benefit from a more explicit discussion of its current limitations.

---

> > > ### Author Response · Authors · 2026-04-06
> > >
> > > Dear Reviewer,
> > >
> > > We appreciate your continued engagement and thoughtful follow-up.
> > >
> > > **Architectural novelty.** We agree that a clearer discussion of how AtomMOF relates to prior work would benefit the paper. We will revise Section 3.3 and add a discussion in the Appendix explicitly stating that: (1) our pairwise representation is inspired by AF3, but replaces the expensive triangle updates $O(N^3d)$ with attention bias $O(N^2d)$; and (2) unlike ADiT and Mofasa, AtomMOF operates directly in structure space without a VAE. We will include the comparison table from our rebuttal in the revised manuscript.
> > >
> > > **MOF-adsorbate experiments.** We appreciate the suggestion. While we agree that a stronger learning-based baseline would be beneficial, adapting existing methods to our task requires many changes in data and modeling assumptions, so would be infeasible within the rebuttal period. We therefore would like to commit to adapting MOFFlow-2, which is the strongest baseline in MOF structure prediction, for this setting within the camera-ready version.
> > >
> > > **Technical novelty.** We appreciate the suggestion to clarify originality and scope.
> > > - *Scaling laws:* While other MOF generation models could potentially exhibit scaling behavior, we expect it to be less favorable due to their architectural complexity (e.g., equivariant IPA, mixed Euclidean/Riemannian input-output spaces). In contrast, our simple all-atom DiT architecture is consistent with scaling behavior demonstrated in related domains (ADiT for crystals, Proteina for proteins). To verify this, we commit to training MOFFlow-2 at smaller (\~40M) and larger scales (\~500M) to directly compare scaling behavior in the camera-ready version.
> > > - *FK steering:* We clarify that adapting FK steering to MOF setting with MLIP guidance is not technically challenging. However, our contribution is in showing that a simple strategy — using off-the-shelf MLIPs as the reward signal — consistently improves structural quality and stability without complex tuning. This contrasts with prior FK steering work that relies on hand-designed energy functions [1] or domain-specific external scoring pipelines [2]. Moreover, this strategy generalizes (1) across MLIPs and (2) even across material domains (MOFs $\to$ inorganic crystals [3]), as shown in the tables below. All hyperparameters match the paper except for Orb, where we reduced the particles from 16 to 8 due to computational limits. We will revise our introduction and appendix to make these contributions more explicit.
> > >
> > > | Steering MLIP | MOF validity | Structure validity | $\Delta E_f$ (eSEN) |
> > > | --- | --- | --- | --- |
> > > | None | 34.16 | 81.70 | 0.126 |
> > > | eSEN | 37.87 (+10.9%) | 91.70 (+12.2%) | 0.020 (−84.1%) |
> > > | UMA | 43.98 (+28.7%) | 94.90 (+16.2%) | 0.017 (−86.5%) |
> > > | Orb | 41.47 (+21.4%) | 95.80 (+17.3%) | 0.036 (−71.4%) |
> > >
> > > | Dataset | Method | Structure validity (%) | $\Delta E_f$ (eV/atom) |
> > > | --- | --- | --- | --- |
> > > | MP-20 | OMatG | 99.26 | 1.4975 |
> > > |  | OMatG + FK | 99.93 (+0.68%) | 0.4074 (−72.8%) |
> > > | MPTS-52 | OMatG | 91.37 | 4.2787 |
> > > |  | OMatG + FK | 99.42 (+8.81%) | 1.3652 (−68.1%) |
> > >
> > > **Limitations.** Thank you for this suggestion. We will add an explicit limitations discussion covering: (1) degraded performance for structures with large atom counts (>500), and (2) insufficient generalization to out-of-distribution structures (e.g., unseen metal nodes or adsorbates). While the vast majority of experimentally synthesized MOFs fall within these bounds, we believe both are important areas for future improvement.
> > >
> > > We sincerely thank the Reviewer for the constructive feedback, which has helped us sharpen the presentation of our contributions and limitations. We will update the manuscript and conduct the follow-up experiments as discussed above for the camera-ready version.
> > >
> > > ---
> > >
> > > [1] Wohlwend et al., "Boltz-1 Democratizing Biomolecular Interaction Modeling," bioRxiv 2024.
> > > [2] Hartman et al., "Controllable protein design through Feynman-Kac steering," arXiv 2025.
> > > [3] Höllmer et al., "Open Materials Generation with Stochastic Interpolants," ICML 2025.

---

### Official Review · Reviewer_3s3X · 2026-03-13

**Soundness:** 4
**Presentation:** 3
**Significance:** 3
**Originality:** 3
**Overall Recommendation:** 6
**Confidence:** 4

**Summary:**

The authors propose AtomMOF, a new model based on diffusion transformers to generate new MOF structures. The model uses Variational Flow Matching and Feynman–Kac steering to achieve high performance in MOF generation relative to the baseline. The authors provide a large set of experiments, including comparisons with other models from the literature, investigation of scaling laws, and a showcase of steering performance for MOFs.

**Compliance With Llm Reviewing Policy:**

Affirmed.

**Key Questions For Authors:**

1) Can you use steering with other models from the baseline, and does it improve stability?
2) Did you consider pre-training your model on a larger dataset of crystalline structures/organic molecules?
3) Did you consider including other baselines like MatterGen in your experiments?

**Limitations:**

yes

**Strengths And Weaknesses:**

The soundness of the submission is strong. The model design seems more than appropriate, and a wide range of experiments validate the design choices and performance compared to other models from the literature.
The presentation remains clear even with the large amount of content in the submission. The context is well presented, and the contributions are clearly stated.
The significance of this article is high, as MOFs are known to be especially challenging to generate and constitute an active and promising research field.
The proposed model also benchmarks many interesting concepts that are rarely used in ML for materials science, such as steering of DiT.

---

> ### Author Rebuttal · Authors · 2026-03-31
>
> Dear Reviewer 3s3X,
>
> We sincerely thank you for your thorough and constructive review. We are grateful for your recognition of the soundness of the paper, the breadth of our experiments, and the significance of our contributions. Below, we address each question in detail.
>
> **Q1. Can steering be applied to baseline models, and does it improve their stability?**
>
> Our MLIP-guided FK steering is applicable to any flow matching model in principle, but requires converting the generative ODE to an SDE. This conversion is straightforward in Euclidean space with Gaussian priors (as in SiT [1]), but non-trivial for our MOF baselines (MOFFlow, MOFFlow-2), which use Riemannian flow matching with non-Gaussian priors on manifolds (e.g., SO(3), tori).
>
> To demonstrate generalizability, we applied our steering to OMatG [2], an inorganic crystal structure prediction model that already operates with an SDE formulation. Specifically, we used the OMatG `Linear-SDE-Gamma` checkpoints for the MP-20 and MPTS-52 datasets with 310 integration steps (OMatG default), guided by UMA (`uma-s-1p2`) with FK hyperparameters identical to our paper setting: 16 particles, $\lambda=2.0$, immediate potential mode, resampling interval 5, and start time 0.8.
>
> | Dataset | Method | Validity (%) $\uparrow$ | $\Delta E_f$ (eV/atom) $\downarrow$ |
> |---|---|---|---|
> | MP-20 | OMatG | 99.26 | 1.4975 |
> | | OMatG + FK | 99.93 (+0.68%) | 0.4074 (−72.8%) |
> | MPTS-52 | OMatG | 91.37 | 4.2787 |
> | | OMatG + FK | 99.42 (+8.81%) | 1.3652 (−68.1%) |
>
> Our approach consistently improves structure validity by 0.68-8.81% and $\Delta E_f$ by 68.10-72.80% across both datasets, confirming that it generalizes across both models and material domains. We appreciate the reviewer's suggestion and will add these results to the Appendix of the revised manuscript.
>
> [1] Ma et al., "SiT: Exploring Flow and Diffusion-based Generative Models with Scalable Interpolant Transformers," ECCV 2024.
> [2] Höllmer et al., "Open Materials Generation with Stochastic Interpolants," ICML 2025.
>
> **Q2. Was pre-training on a larger dataset of crystals or organic molecules considered?**
>
> Thank you for this interesting question. We did not explore pre-training in the current work, but consider it a highly promising future direction. Recent universal MLIPs (e.g., UMA [3]) have demonstrated that physical knowledge transfers effectively across inorganic crystals, organic molecules, and MOFs, and we believe this extends naturally to the generative domain. Moreover, our scaling law analysis shows that AtomMOF's performance improves predictably with model capacity without saturation, suggesting that the combination of a larger model with broader pre-training data is particularly well-suited to our framework. We will note this as an important future direction in the revised manuscript!
>
> [3] Wood et al., "UMA: A Family of Universal Models for Atoms," NeurIPS 2025.
>
> **Q3. Was MatterGen considered as a baseline?**
>
> Thank you for the suggestion. We did not include MatterGen as a baseline due to differences in system scale and material class; MatterGen was trained on inorganic crystals with a maximum of 20 atoms per unit cell, whereas AtomMOF was trained on MOFs with up to ~1,300 atoms per unit cell. This difference makes a direct comparison difficult. That said, as discussed in Q2, pre-training on broader crystal datasets is a promising future direction for our framework, and comparing with MatterGen in such a unified setting would be a natural next step.

---

### Official Review · Reviewer_R6RE · 2026-03-15

**Soundness:** 2
**Presentation:** 3
**Significance:** 3
**Originality:** 2
**Overall Recommendation:** 3
**Confidence:** 3

**Summary:**

This paper proposes AtomMOF, an all-atom flow-matching model based on a Diffusion Transformer for predicting equilibrium 3D structures of MOFs and MOF-adsorbate systems from 2D building-block graphs and adsorbate identity. Unlike prior MOF generation methods that use coarse-grained or geometry-constrained parameterizations, the proposed model predicts atom coordinates and lattice parameters in an unconstrained all-atom space. The paper also studies scaling behavior across model sizes and introduces MLIP-guided Feynman-Kac steering to improve validity and stability during sampling. Empirically, AtomMOF improves structure matching and RMSD on the BW benchmark, improves property preservation relative to prior MOF baselines, and on ODAC25 shows much faster recovery of adsorption configurations than GCMC/random sampling while sometimes identifying candidates with lower adsorption energy than the reference set.

**Compliance With Llm Reviewing Policy:**

Affirmed.

**Key Questions For Authors:**

- The steering and adsorption-energy results appear to rely heavily on the same MLIP family for guidance, relaxation, and/or evaluation. Could the authors report a validation with an independent oracle (e.g., a different MLIP or a DFT subset) to show that the gains are not primarily due to metric-model alignment? A positive answer would substantially increase my confidence in the physical validity of the conclusions.

- For ODAC25, the paper uses the public validation split as a test set because the official test set is unavailable. Could the authors clarify exactly which hyperparameters were selected on which split, and what safeguards were used to avoid tuning on the reported evaluation set? A clearer answer would strengthen the soundness of the empirical protocol.

- The treatment of metal-node reference coordinates during ODAC25 inference is important but currently somewhat under-explained. When nodes are indexed by SMILES and a specific structure is sampled at inference time, are those templates drawn strictly from training data? Also, how often do structurally identical or near-identical nodes appear across splits? Clarifying this would help assess how de novo the prediction setting is.

- In Section 5.1, GCMC and random sampling are evaluated on the framework predicted by AtomMOF, which makes the comparison primarily about adsorbate placement rather than full-system generation. Could the authors discuss this choice more explicitly, and if possible provide an additional comparison that better isolates end-to-end generation quality?

- What are the dominant failure modes for the unmatched BW structures and unsuccessful ODAC25 relaxations? A more granular breakdown (e.g., lattice mismatch, local coordination failure, atomic overlap, or adsorption-site diversity) would make the method easier to interpret and build upon.

Overall, if the authors could carefully address my above concerns, I would like to raise my score.

**Strengths And Weaknesses:**

**Strengths**. The paper addresses an important AI-for-science problem at the intersection of generative modeling and materials discovery. The move from rigid or coarse-grained MOF generation to fully all-atom prediction is meaningful, especially because host-guest interactions can materially affect downstream adsorption behavior. The formulation is technically interesting: the model jointly predicts atomic coordinates and lattice parameters in a flexible flow-matching framework, and the MLIP-guided Feynman-Kac steering component is a sensible way to improve sample validity in an unconstrained generation space. The empirical evaluation is broad. On BW, AtomMOF consistently improves over MOFFlow, MOF-BFN, and MOFFlow-2 under both strict and loose StructureMatcher thresholds, and it also preserves pore/property statistics better than the strongest baseline. The ODAC25 experiments are also compelling from an application standpoint: the model recovers adsorption configurations much faster than GCMC and can produce initializations that sometimes lead to lower adsorption energies than the dataset reference. Finally, the paper is reasonably clear overall, and the availability of code, data, and released checkpoints substantially improves reproducibility.

**Weaknesses**. First, the absolute structure-matching performance is still modest under the strict evaluation protocol, so the method is not yet close to robust de novo structure prediction in an absolute sense. Second, some of the strongest physical claims rely on MLIP-based guidance and MLIP-based evaluation; this makes it unclear how well the gains transfer to an independent oracle such as DFT or even a different potential. Third, for ODAC25 the paper uses the public validation split as a test set because the official test set is unavailable; this is understandable, but it weakens the conclusiveness of the empirical protocol. Fourth, in the adsorption-configuration comparison, GCMC and random sampling are evaluated on the framework predicted by AtomMOF, which isolates adsorbate placement rather than comparing full end-to-end system generation. Finally, the handling of metal-node reference coordinates/templates during ODAC25 inference deserves clearer discussion, since this detail matters for judging how de novo the prediction setting really is.

---

> ### Author Rebuttal · Authors · 2026-03-31
>
> Dear Reviewer R6RE,
>
> We thank the reviewer for the detailed review. Below, we address each concern in detail. **Tabular results are available [HERE](https://anonymous.4open.science/r/AtomMOF-Rebuttal-425C).**
>
> **W1. Absolute structure-matching performance remains modest under the strict protocol.**
>
> We appreciate this observation and provide context. We believe our absolute performance is strong given the task difficulty. The community standard in CSP is stol=0.5; stol=0.3 is a stricter complementary measure we report, where lower absolute values are expected. Under stol=0.5, AtomMOF-L achieves a 29.96% match rate with a single sample, surpassing the inorganic CSP SOTA of 27.38% by OMatG [1] on MPTS-52. This is notable since MOFs average \~150 atoms, an order of magnitude larger than MPTS-52 (~18 atoms). Our scaling law analysis further suggests continued improvement with increased model capacity.
>
> [1] Höllmer et al., "Open Materials Generation with Stochastic Interpolants," ICML 2025.
>
> **W2/Q1. Claims rely on the same MLIP for steering and evaluation; gains may reflect metric-model alignment rather than true physical improvement.**
>
> Thank you for raising this concern. To rule out metric-model alignment, we conducted cross-MLIP experiments for both experiments.
>
> *Steering experiment (Table 1).* We steered with three independent MLIPs: eSEN, UMA, and Orb (detailed settings are in table captions). Notably, steering with UMA and Orb (different from eSEN evaluator) yields higher validity scores than steering with eSEN itself. This confirms that the improvements reflect genuine structural quality rather than overfitting to a specific potential.
>
> *Adsorption energy experiment (Table 2).* We kept eSEN for relaxation, but evaluated adsorption energy with both eSEN and UMA. Success rates and energy statistics are consistent across evaluators, further ruling out metric-model alignment. We will include these results in the Appendix of the revised manuscript.
>
> **W3/Q2. ODAC25 uses the validation split as test set; which hyperparameters were selected on which split, and what safeguards prevent tuning on the evaluation set?**
>
> We appreciate this concern. All model hyperparameters were developed on the BWDB and transferred to ODAC25 without modification (Appendix C). The only difference is the maximum atom count per curriculum stage, set a priori from dataset statistics: 200 atoms covers \~83.5% of BW but only \~47% of ODAC25, so we used 300 to increase coverage (~74.6%). To monitor training, we held out 3,130 samples from the training split for sanity checks (e.g., confirming validation loss decreases). The official validation split was used solely for final evaluation and was never tuned against.
>
> **W4/Q4. GCMC/RS comparison isolates adsorbate placement rather than end-to-end generation quality.**
>
> End-to-end comparison with GCMC/RS is difficult because they cannot predict MOF structures; providing them with ground-truth frameworks gives them an unfair advantage. Comparison against a full traditional pipeline (e.g., traditional CSP + GCMC/RS) is also not meaningful, as traditional CSP methods (e.g., RS/EA) fail on MOFs within a reasonable computational budget (\~0% match rate). We therefore isolate adsorbate placement as the fairest comparison.
>
> **W5/Q3. Metal-node template handling at ODAC25 inference is under-explained. Are templates strictly drawn from the training set? How often do identical nodes appear across splits?**
>
> Metal-node templates are drawn strictly from the training set and serve only as coordinate initialization; the model freely predicts all-atom coordinates and adjusts node geometry through interaction with linkers and adsorbates. Specifically, we construct a dictionary mapping each node SMILES to a list of all corresponding node structures from training data (883 unique nodes). At inference, we randomly sample one structure per SMILES as initialization. Nearly all val/test nodes appear in this dictionary (472/473 and 128/136); for rare absent cases, we initialize the structures from a Gaussian distribution.
>
> **Q5. What are the dominant failure modes for unmatched BW structures and unsuccessful ODAC25 relaxations?**
>
> *BW match failure (Table 3)*. The dominant failures are coordinate-level, not lattice-level (only 16.5% from lattice mismatch). Primary failure modes are atomic overlaps and local coordination errors (overcoordinated C/H/N). Notably, 16.4% of unmatched structures pass all MOFChecker checks, suggesting valid alternative configurations.
>
> *ODAC25 relaxation failure (Table 4)*. Failures are driven by framework quality, not adsorption-site diversity (7.52Å vs. 7.43Å). The dominant signal is atomic overlaps (0% in successes vs. 32.9% in failures), followed by undercoordinated metals (4.5% vs. 57.9%).
>
> Both analyses motivate our FK steering, which targets these failure modes by improving structural quality (W2/Q1).

---

### Official Review · Reviewer_PTaP · 2026-03-15

**Soundness:** 3
**Presentation:** 3
**Significance:** 4
**Originality:** 4
**Overall Recommendation:** 6
**Confidence:** 5

**Summary:**

This paper studies the problem of predicting the full atomic structure of metal–organic frameworks (MOFs). MOF structure prediction is challenging because of their large system size, modular composition, and flexible organic linkers. Existing approaches often rely on rigid building block assumptions where bond lengths and angles are fixed during generation.
The paper proposes ATOMMOF, an all-atom generative model that directly predicts atomic coordinates of MOF–adsorbate systems. The model is built on a diffusion transformer architecture that jointly models atomic coordinates and lattice parameters. Unlike prior methods, the approach does not rely on rigid-body assumptions for linkers and instead models the full internal geometry of the structure. The architecture uses atom-level features, pairwise geometric representations, and transformer-based interaction blocks to capture long-range structural dependencies.
To improve sampling quality, the method further integrates a machine learning interatomic potential with a steering mechanism during generation. Experiments on MOF structure prediction benchmarks show consistent improvements over existing optimization-based and generative baselines. The proposed model achieves higher match rates and lower RMSD, particularly under stricter structural matching criteria.
Overall the paper explores a promising direction toward fully flexible generative modeling of MOF structures.

**Compliance With Llm Reviewing Policy:**

Affirmed.

**Ethical Review Concerns:**

My concerns has been fully addressed and after considering all the experiments and additional experiments addressed in other reviewer's comment, I will increase my original score to 6. As a researcher working on AI for material science, particularly with MoF, this is meaningful contribution for scientific community.

**Final Justification:**

My concerns has been fully addressed and after considering all the experiments and additional experiments addressed in other reviewer's comment, I will increase my original score to 6. As a researcher working on AI for material science, particularly with MoF, this is meaningful contribution for scientific community.

**Key Questions For Authors:**

1) How sensitive is the generation quality to the choice of MLIP used in the steering procedure?
2) Does the model maintain good performance when predicting larger MOF structures than those seen during training?
3) Have the authors evaluated the diversity or novelty of generated structures beyond reconstruction accuracy?

**Limitations:**

yes

**Strengths And Weaknesses:**

Pros:
- The paper addresses an important problem in computational materials science. Accurate and scalable prediction of MOF structures is highly relevant for applications such as gas storage, catalysis, and carbon capture.
- MOFs are especially important for large-scale materials discovery pipelines, where efficient structure prediction can significantly accelerate screening of candidate frameworks.
- The proposed all-atom formulation is a meaningful step beyond previous rigid-body MOF generation approaches. Modeling internal bond geometry directly allows the method to capture structural flexibility that earlier models ignore.
- The architecture design is reasonable. Using a transformer-based generative model with atom-level and pairwise features provides a natural way to model long-range interactions in crystal structures.
- Integrating MLIP-based steering during sampling is a practical design choice that improves structure quality without significantly increasing complexity.
- Experimental results are convincing. The method consistently outperforms previous MOF-specific baselines as well as general crystal structure prediction methods.
- Improvements are particularly noticeable under stricter structure matching thresholds, suggesting the all-atom modeling is indeed capturing important geometric details.
- The scaling results are also interesting, showing that performance improves predictably with model size.

Cons:
- The evaluation focuses primarily on structure reconstruction accuracy. Additional analysis of generated structure diversity or novelty would further strengthen the generative modeling claims.
- The MLIP steering component is interesting but somewhat briefly described. More details on its stability or sensitivity to different potentials would improve clarity.
- Some architectural choices (for example feature initialization from building blocks) could benefit from a bit more intuition or ablation discussion.

---

> ### Author Rebuttal · Authors · 2026-03-31
>
> Dear Reviewer PTaP,
>
> We sincerely thank you for your thorough and constructive review. We are grateful for your positive assessment of our work. Below, we address each concern in detail.
>
> **W1/Q3. Metrics focus on reconstruction accuracy. Have the authors considered evaluating diversity and novelty?**
>
> Thank you for raising this point. Since our task is structure prediction, we prioritize reconstruction fidelity measures (match rate, RMSD), consistent with analogous structure prediction tasks [1,2]. To verify whether diversity/novelty metrics are applicable to our setting, we analyzed the BW test set using MOFid [3]: among the 19,792 structures, systems sharing identical building block SMILES exhibit zero topology diversity. This indicates that the BW dataset contains no polymorphic variation, making diversity/novelty metrics uninformative for this benchmark.
>
> That said, we agree that diversity and novelty are meaningful metrics in settings where multiple valid structures exist for the same input, such as polymorphic MOFs. Exploring novelty and diversity under this setting is an interesting direction we plan to investigate in future work.
>
> [1] Jiao et al., "Crystal Structure Prediction by Joint Equivariant Diffusion," NeurIPS 2023.
> [2] Höllmer et al., "Open Materials Generation with Stochastic Interpolants," ICML 2025.
> [3] Bucior et al., "Identification Schemes for Metal–Organic Frameworks To Enable Rapid Search and Cheminformatics Analysis," Crystal Growth & Design 2019.
>
>
> **W2/Q1. MLIP-based steering is interesting but briefly mentioned. More details on its stability or sensitivity to different potentials would improve clarity.**
>
> Thank you for this excellent suggestion! We conducted additional ablations replacing the eSEN potential with two alternative MLIPs: UMA (`uma-s-1p2`) and Orb (`orb-v3-conservative-inf-omat`). All settings follow those used for eSEN, except that for Orb we reduced the number of FK particles from 16 to 8 due to computational constraints.
>
> FK steering yields consistent and substantial improvements across all three MLIPs — MOF validity improves by 10.9-28.7%, structure validity improves by 12.2-17.3%, and $\Delta E_f$ drops by 71.4-86.5% relative to the unsteered baseline — demonstrating that our strategy is robust to the choice of MLIP. Notably, UMA and Orb even outperform eSEN on validity metrics, suggesting that our framework can readily benefit from future advances in universal MLIPs. These results will be added to Appendix E of the revised manuscript.
>
> | Method | FK | MOF validity (%) | Structure validity (%) | eSEN $\Delta E_f$ (eV/atom) |
> |---|---|---|---|---|
> | No steering | — | 34.16 | 81.70 | 0.126 |
> | eSEN steering | ✓ | 37.87 (+10.9%) | 91.70 (+12.2%) | 0.020 (−84.1%) |
> | UMA steering | ✓ | 43.98 (+28.7%) | 94.90 (+16.2%) | 0.017 (−86.5%) |
> | Orb steering | ✓ | 41.47 (+21.4%) | 95.80 (+17.3%) | 0.036 (−71.4%) |
>
> **W3. Some architectural choices (e.g., feature initialization from building blocks) could benefit from more intuition or ablation discussion.**
>
> Thank you for this suggestion. We conducted additional ablations on two key components of our building block feature initialization: pairwise representation and bond topology in organic linkers. To keep the experiment tractable within the rebuttal period, we trained AtomMOF-M for 500 epochs with the maximum atom count limited to 200 (~83.5% of training data); all other settings are identical to the paper.
>
> | Configuration | MR (stol=0.3) | MR (stol=0.5) |
> |---|---|---|
> | Full | 11.18 | 30.80 |
> | No pairwise repr. | 8.29 (−25.8%) | 26.60 (−13.6%) |
> | No bond topology | 10.30 (−7.9%) | 30.07 (−2.4%) |
>
> The pairwise representation contributes most significantly to prediction quality. This is intuitive: structure prediction involves placing atoms at correct distances from one another, and explicit pairwise representations allow the model to reason directly about interatomic distances rather than inferring them from single-atom features alone. Bond topology has a smaller effect, likely because connectivity information is partially recoverable from atom types and pairwise distances. We will include this result in the Appendix of the revised manuscript.
>
> **Q2. Does the model generalize to MOF structures larger than those in training?**
>
> Currently, generalization beyond \~1,300 atoms (the maximum in our training dataset) is out of reach, and we consider scaling to larger structures an important future direction. That said, the vast majority of experimentally synthesized MOFs fall within \~500 atoms per unit cell; MOFDiff explicitly limits structure size for this reason, and prior MOF generation works (ADiT, Zatom-1, Mofasa) all restrict to $\leq200$ atoms. Within this practically relevant range, our model achieves strong performance ~33.8% match rate for $\leq200$ atoms for 1 sample generation).

---

> > ### Author Rebuttal · Reviewer_PTaP · 2026-04-05
> >
> > My concerns have been fully addressed. After carefully considering the original submission as well as the additional experiments provided in response to other reviewers’ comments, I will increase my score to 6.
> >
> > As a researcher working on AI for materials science, particularly in the context of MOFs, I find this work to be a meaningful contribution to the scientific community. I hope it can further accelerate progress toward closed-loop materials discovery for MOFs.
> >
> > One minor suggestion for future work is to provide a more detailed discussion of the SevenNet-Omni model as an MLIP, given its demonstrated effectiveness for MOF-related tasks.

---

> > > ### Author Response · Authors · 2026-04-06
> > >
> > > We sincerely thank the Reviewer for the generous reassessment and kind words. We share the hope that this work contributes to the scientific community and materials discovery for MOFs.
> > >
> > > Thank you also for the suggestion regarding SevenNet-Omni. We ran FK steering with SevenNet-Omni (task `odac23`) using the same hyperparameters, except the number of particles, which we reduced from 16 to 8 due to limited computational budget (as with Orb). SevenNet steering yields improvements consistent with the other MLIPs (+21.5% MOF validity, +16.9% structure validity, −62.7% $\Delta E_f$), further reinforcing the generalizability of our approach. We will add SevenNet to the revised manuscript.
> > >
> > > | Method | FK | MOF validity (%) | Structure validity (%) | eSEN $\Delta E_f$ (eV/atom) |
> > > |---|---|---|---|---|
> > > | No steering | — | 34.16 | 81.70 | 0.126 |
> > > | eSEN steering | ✓ | 37.87 (+10.9%) | 91.70 (+12.2%) | 0.020 (−84.1%) |
> > > | UMA steering | ✓ | 43.98 (+28.7%) | 94.90 (+16.2%) | 0.017 (−86.5%) |
> > > | Orb steering | ✓ | 41.47 (+21.4%) | 95.80 (+17.3%) | 0.036 (−71.4%) |
> > > | SevenNet steering | ✓ | 41.52 (+21.5%) | 95.50 (+16.9%) | 0.047 (−62.7%) |

---

### Decision · Program_Chairs · 2026-04-30

**Decision:**

Reject

**Comment:**

The paper presents a flow-based model, AtomMOF, for generating equilibrium MOF-adsorbate structures. In contrast to prior MOF generation methods that rely on coarse-grained representations with fixed bond lengths and angles, AtomMOF operates in an unconstrained all-atom space, directly predicting atom coordinates and lattice parameters while explicitly modeling MOF-adsorbate interactions. The paper also studies scaling behavior across model sizes and introduces MLIP-guided Feynman-Kac steering to improve sampling validity and stability. Experimental results on standard benchmarks show improved reconstruction accuracy over existing methods and greater sample efficiency than GCMC/random sampling baselines for recovering adsorption configurations.

The overall formulation seems reasonable, and the problem is important. Reviewers generally found the empirical performance promising, while also asking for further clarification in several aspects of the evaluation and the technical positioning of the work.

* Reliability of the MLIP-based steering and evaluation, and consistency with prior evaluation protocols, as asked by Reviewers R6RE and 9M86. This is a reasonable concern, since the proposed Feynman-Kac guidance depends on pretrained MLIPs whose generalization to generated structures is not guaranteed, and the same issue also affects parts of the evaluation. The rebuttal was helpful in showing that FK steering yields consistent gains under alternative MLIPs, such as UMA and Orb, which is supportive. Reviewer 9M86 also raised a concern that the evaluation relies on MLIPs rather than DFT as in some prior works, which may reduce the reliability of the conclusions. The authors explained that full DFT-based evaluation is substantially more expensive in their setting, which is understandable. Nevertheless, Nevertheless, I think the paper would be substantially stronger with representative DFT validation on a subset.
* Data split and possible leakage concerns on ODAC25, as raised by Reviewer R6RE. The reviewer asked whether the evaluation setting may have been tuned in a way that risks leakage. The authors’ clarification about the split construction and evaluation protocol appears reasonable to me and largely addresses this concern.
* Evaluation beyond reconstruction accuracy, especially structural diversity and novelty, as asked by Reviewer PTaP. The authors responded that the BW test set contains essentially no polymorphic variation, making diversity and novelty metrics less informative in that setting. This seems reasonable to me, although I still hope the authors can clarify this limitation more explicitly in the final version.
* Technical novelty and the presentation of the main methodological contribution, especially as raised by Reviewer 9M86. I agree that the originality and impact of the proposed techniques could be articulated more clearly. In particular, it remains somewhat unclear which component should be viewed as the main conceptual advance, and to what extent the observed gains arise from a genuinely new modeling idea versus a careful integration of existing ingredients. The rebuttal on this point did not fully convince me; for example, the paper does not directly establish whether the reported scaling behavior is distinctive to AtomMOF or reflects a more general trend.

Overall, while the paper addresses an important problem and shows promising empirical results, I am not yet sufficiently convinced by the current version on two central points: the reliability of conclusions that depend heavily on MLIP-based guidance and evaluation, and the clarity of the technical novelty beyond a strong systems-level integration of existing ingredients. The rebuttal addressed part of the concerns, particularly by showing that the FK steering gains are consistent across alternative MLIPs, but in my view this is not yet enough to fully resolve the main questions about validation and technical contribution. For these reasons, I do not support acceptance in the current form, although I believe the work is promising and could become a stronger submission with clearer positioning and more direct validation.